# Weakly Supervised Evaluation of Multi-Turn Objective-Driven Interactions

## Abstract

Large language models (LLMs) have seen increasing popularity in enterprise applications where AI agents and humans engage in objective-driven interactions. However, these systems are difficult to evaluate: data may be complex and unlabeled; human annotation is often impractical at scale; custom metrics can monitor for specific errors, but not previously-undetected ones; and LLM judges can produce unreliable results. We introduce the first set of weakly supervised metrics for objective-driven interactions, leveraging statistical properties of unlabeled interaction data and reasonable assumptions on the performance of frontier LLMs. We apply small fine-tuned LLMs to adapt to distributional shifts and develop metrics for clustering user goals, classifying goal completion, and quantifying LLM uncertainty without grounding evaluations in human-generated ideal responses. Our approach is validated on open-domain and task-specific interaction data.

## 1 Introduction

LLMs are seeing increasing use in business applications such as task-oriented dialogue (TOD), agentic systems that assist with work tasks, and customer service systems. Despite the prevalence of such objective-driven conversational systems, the development of evaluation tools to measure their performance has lagged behind, primarily relying on LLM judges or custom metrics to detect specific conversational attributes. Additional challenges are introduced by distributional shifts in data when LLMs are prompted or fine-tuned to specialize in narrow application domains, or when agents interact with each other, reason, use tools, and modify shared environments such as an IDE or document editor. These systems produce long, complex interactions – a weakness of even the most capable LLMs (Laban et al., 2025). We introduce three LLM judge-free metrics for labeling user goals, goal completion, and LLM uncertainty. Our metrics are designed for frontier LLMs where errors are rare and conversations are typically of high quality. We leverage these assumptions and implicit properties of objective-driven interaction data to introduce the first set of weakly supervised metrics for objective-driven interactions. Specifically, we present: a stable clustering algorithm for identifying user goals; the use of small fine-tuned LLMs to adapt to distributional shifts; and a novel *response tree* structure representing multiple possible LLM responses, which can be efficiently constructed and used to understand the conditional distribution of LLM responses given a specific prompt. We apply these tactics to develop metrics for labeling user goals, measuring goal completion, and quantifying LLM uncertainty: evaluation axes of interest to agentic system developers.

## 2 Prior Work

The first metrics for computer-generated text relied on human reference answers for comparison. ROUGE (Lin, 2004) compares n-gram recall between generated and ideal summaries, and BLEU (Papineni et al., 2002), originally proposed to evaluate machine translation, compares n-gram precision. More complex approaches account for semantic similarity (Denkowski & Lavie, 2014; Zhang et al., 2019), but still require reference answers and can miss details that significantly change the meaning of text (Saadany & Orăsan, 2021). Researchers have also sought to rate machine-generated text on numeric or binary scales, as these evaluations are useful for LLM alignment. Often, these ratings are produced using human annotation. For example, HelpSteer (Wang et al., 2023) employed 200 human annotators to produce a dataset of 37,120 prompt-response pairs evaluated on five axes.

HelpSteer3 employed over 7,000 annotators to produce textual feedback on 40,500 multi-turn conversations (Wang et al., 2025).

The difficulty of scaling metrics based on human evaluators, and the development of highly capable large language models, has led to the use of LLM-based metrics. Perplexity measures the LLM's confidence in predicting the next token, with high perplexity corresponding to less probable tokens. Many evaluators use LLM-as-a-judge: evaluating the output of one LLM by prompting another (Gu et al., 2024). LLM judge frameworks include G-EVAL (Liu et al., 2023b), which uses chain-of-thought reasoning, and frameworks that combine decision trees with LLM judgments, breaking the evaluation task into smaller steps (DeepEval, 2025b). LLM-as-a-judge has been applied to multi-turn conversations (Guan et al., 2025; Wang et al., 2024), where LLMs are known to suffer from new challenges as the length of conversations increase (Laban et al., 2025). However, LLM judges are known to be unreliable (Li et al., 2025) and highly sensitive to the design of evaluation methods (Baumann et al., 2025). Finally, researchers have identified position bias (Zheng et al., 2023), verbosity bias (Saito et al., 2023), familiarity bias (a preference for outputs with lower perplexity) (Stureborg et al., 2024), inconsistent outputs (Stureborg et al., 2024; Wei et al., 2025), and sensitivity to wording changes in prompts (Wei et al., 2025) as potential challenges for LLM judges.

Fine-tuning has also been applied for text evaluation: for example, in the computational social sciences, (Carammia et al., 2024) show that small fine-tuned models can match the performance of large proprietary models on text classification tasks and (Cao et al., 2025) applies LLMs for simulating distributions of population-level survey responses. However, previous fine-tuning approaches primarily retain the LLM-as-a-judge paradigm (Zhu et al., 2025).

## 3 PROBLEM STATEMENT

We define a multi-turn objective-driven interaction as an interaction between a human user and one or more LLMs. Each turn $i$ consists of a user prompt $p_i$ and an LLM response $r_i$, both represented as text strings. The user has a latent *goal* $g$ which they seek to complete with the assistance of the LLM(s). We seek to infer three main attributes of the interaction, selected because they are of practical interest to AI system developers [1].

- The latent goal $g$.

- Whether $g$ was completed during the conversation.

- The LLM's uncertainty when producing a response, as measured by the presence of multiple different answers in the response tree (4.3): an approximation of the conditional distribution of responses for a given prompt.

Prior work on clustering user goals and classifying their completion primarily used LLM judges. DeepEval, a popular package for multiple LLM metrics, uses LLMs to first extract a list of user intents, then determine how many intents are completed during the conversation (DeepEval, 2025a). The fraction *# completed intents/# intents* is reported as the completion percentage. Other industry solutions follow similar patterns (AI, 2025a). Our approach to classify completion has more in common with unsupervised anomaly detection, which has previously been implemented on short text strings using text embedding models and mixture distributions (Ait-Saada & Nadif, 2023) or clustering algorithms (Wang et al., 2022). Completion labeling by modeling the distribution of completed responses is one of our novel contributions.

Prior work on LLM uncertainty has proposed a variety of definitions and measurement tactics (Shorinwa et al., 2025). Log probabilities carry uncertainty information, with some libraries taking this approach (AI, 2025b). Other approaches include semantic entropy, which samples multiple responses (Kuhn et al., 2023; Bouchard et al., 2025), conformal prediction to estimate uncertainty in multiple-choice question answering (Ye et al., 2024), and training an LLM to estimate uncertainty using a dataset of prompts with known ideal answers (Liu et al., 2024).

---

[1]Our software implementation of these metrics will be available in the final version of this paper.

## 4 METHODOLOGY

We make the following assumptions on the quality of interactions.

- The user always has one goal to complete: for example, the user could be speaking with an LLM customer service assistant or using an agent to complete a work task. If the user has multiple goals, they should be separated into multiple interactions.

- In a majority of the interactions, the user completes their goal. We justify this by noting that if an LLM produces frequent failures, they can be detected by inspection. Our method applies to settings where failures are rare, thus difficult to detect by inspection.

### 4.1 WEAKLY SUPERVISED CLUSTERING OF HUMAN-LLM CONVERSATIONS

Extracting insights from unlabeled data is often more difficult than it initially appears. Some evaluation suites use LLMs themselves to label the user's goal, although this can be unstable (Stureborg et al., 2024). When the labels are known, prompting the judge LLM to assign more than one label can stabilize the results (Guerdan et al., 2025), however when the labels are unknown, prompting the LLM to generate new labels produces different results when the dataset is shuffled. On the opposite end of the spectrum, unsupervised clustering algorithms such as k-means can produce stable clusters of vector data. To cluster text using such methods, one could use an embedding model (Neelakantan et al., 2022). The input to an embedding model is a sequence of tokens and the output is a vector $v$ of fixed size. The output $v$ is deterministic (excluding the effects of floating-point precision and batch sizes) and has the property that if $s_1$ and $s_2$ are semantically similar, $v_1$ and $v_2$ have high cosine similarity. However, clusters produced via k-means lack interpretable labels and one must know the number of clusters in advance, which is not realistic for our use case.

We introduce **LLM-guided clustering**: an algorithm 1 that combines k-means with the language understanding capabilities of LLMs to discover an *unspecified number* of labeled clusters from unlabeled interactions. The algorithm proceeds in three phases.

We preprocess the data by prompting an LLM to summarize the user's goal in each interaction as free-form text summaries, denoted $s_1, \ldots, s_N$. For specialized applications, a short context paragraph improves the quality of these summaries; examples of both context and summaries are provided in Appendix B. We embed $s_1, \ldots, s_n$ using OpenAI's `text-embedding-3-small` model, which outputs vectors $v_1, \ldots, v_n \in \mathbb{R}^{1536}$.

In Phase 1 we generate $k_1$ clusters using k-means on $v_1, \ldots, v_n$. (The value of $k_1$ should be an overestimate of the number of clusters actually present.) To label these clusters, we prompt an LLM to describe each cluster, given ten randomly sampled summaries that are *in* the cluster and ten summaries that are *not in* the cluster. The use of positive and negative examples produces higher quality descriptions. The output of Phase 1 is $k_1$ clusters with text descriptions $L_1, \ldots, L_{k_1}$. We embed each description to produce vectors $\boldsymbol{d}_1, \ldots, \boldsymbol{d}_{k_1}$.

In Phase 2 we use the LLM to iteratively merge clusters. We first construct a distance matrix $D \in \mathbb{R}^{k_1 \times k_1}$ where $D_{ij} = (\boldsymbol{d}_i^\top \boldsymbol{d}_j)/(\|\boldsymbol{d}_i\|_2 \|\boldsymbol{d}_j\|_2)$; the cosine similarity of $\boldsymbol{d}_i$ and $\boldsymbol{d}_j$. Using $D$, we iteratively select the largest value of $D_{ij}$ and prompt the LLM to determine whether clusters $i$ and $j$ should be merged, using ten randomly sampled summaries each from clusters $i$ and $j$ and ten sampled from other clusters. This is the **only** point at which we rely on an LLM to make a clustering decision. We terminate the algorithm after all current clusters have failed to merge. This algorithm outputs labels $\{a_1, \ldots, a_n\}$ and text descriptions $L_1, \ldots, L_k$ for $k \leq k_1$ clusters.

### 4.2 INTERACTION COMPLETENESS

In this section, we seek to discover whether the user's interaction with the LLM is complete; that is, whether the LLM correctly and fully addressed the user's goal. Consider a dataset of complete multi-turn conversations from distribution $D$. We define a new distribution $D'$ where the last response in each conversation is augmented with a special `end` tag and a function $\text{LLM}_{D'}$, defined such that $\text{LLM}_{D'}(\boldsymbol{p})$ returns the most likely token sequence $\boldsymbol{r}$ to follow prompt $\boldsymbol{p}$ under distribution $D'$. (A full definition of $\text{LLM}_{D'}$ is provided in Appendix A.) Then if $c = [\boldsymbol{p}_1, \boldsymbol{r}_1, \ldots, \boldsymbol{p}_n, \boldsymbol{r}_n]$ is a full

**Data:** Conversations $C = \{c_1, \ldots, c_n\}$
**Data:** Maximum number of clusters $k_1$
**Result:** Number of clusters $k$
**Result:** Text labels of clusters $L = \{l_1, \ldots, l_k\}$
**Result:** Cluster assignments $A = \{a_1, \ldots, a_n\}$ where $a_i \in \{1, \ldots, k\}$
$S \leftarrow \{\text{SUMMARIZE}(c_i) \mid c_i \in C\}$ ;                     /* Preprocess data */
$V \leftarrow \{\text{EMBED}(s_i) \mid s_i \in S\}$;
$A \leftarrow \text{KMEANS}(V, k_1)$ ;                          /* Initialize clusters */
$L_i \leftarrow \texttt{null}, i = 1, \ldots, k_1$;
$i \leftarrow 0$;
**while** $i < k_1$ **do**
    $\text{cluster} \leftarrow \{c_k \mid a_k = i, k = 1, \ldots, n\}$;
    $L_i \leftarrow \text{LLM}(\text{Describe cluster}_i)$ ;                 /* Label initial clusters */
    $i \leftarrow i + 1$;
**end**
$D \leftarrow \text{SIM}(\{\text{EMBED}(L_i) \mid L_i \in L\})$ ;    /* $D_{ij}$ = cosine similarity of $L_i, L_j$ */
$\text{failures} \leftarrow 0$ **while** *failures* $< |L|$ **do**
    $i, j \leftarrow \arg\max(D)$;
    $\text{cluster}_i \leftarrow \{c_k \mid a_k = i, k = 1, \ldots, n\}$;
    $\text{cluster}_j \leftarrow \{c_k \mid a_k = j, k = 1, \ldots, n\}$;
    **if** $\text{LLM}$(*Should cluster$_i$ and cluster$_j$ be merged?*) *is True* **then**
        $A \leftarrow \text{MERGE}(A, i, j)$ ;                     /* Merge similar clusters */
        $L_i \leftarrow \text{LLM}(\text{Describe } \text{CONCAT}(\text{cluster}_i, \text{cluster}_j))$ ;   /* Label merged cluster */
        $\text{DELETE}(L_j)$ ;                          /* Clean up after merge */
        $D_{j,:} \leftarrow -\infty$;
        $D_{:,j} \leftarrow -\infty$;
        $\text{failures} \leftarrow 0$;
    **else**
        $D_{i,j} \leftarrow -\infty$;
        $\text{failures} \leftarrow \text{failures} + 1$;
    **end**
**end**

**Algorithm 1:** Weakly Supervised Clustering of Human-LLM Conversations

interaction, we define

$$P_{D'}(\text{end} \mid c) = P(\text{LLM}_{D'}(\text{CONCAT}(\boldsymbol{p}_1, \boldsymbol{r}_1, \ldots, \boldsymbol{p}_n, \boldsymbol{r}_n)) = \text{end}).$$

This is the probability that, $c$ is used as the prompt, the next response token is end. If $c' = [\boldsymbol{p}_1, \boldsymbol{r}_1, \ldots, \boldsymbol{p}_k, \boldsymbol{r}_k], k < n$ is an *incomplete* interaction and

$$P_{D'}(\text{end} \mid c) > P_{D'}(\text{end} \mid c').$$

Thus in expectation, $\text{LLM}_{D'}$ correctly labels conversations as complete or incomplete.

In practice, we consider two types of distributions. The *base chat distribution* represents unstructured human-LLM chats similar to the data found in the pretraining datasets used to train modern LLMs (Grattafiori et al., 2024). Objective-driven systems may deviate from the base chat distribution by introducing reasoning, tool use, interactions with a shared environment such as an IDE, and multiple agents. We approximate these distributional shifts by training a LoRA adapter (Hu et al., 2021) for a LLaMA3.2-8B completion model (Grattafiori et al., 2024) to approximate $D'$ using a tagged, unlabeled dataset where the majority of interactions are complete (thus incomplete interactions are outliers). We train the model to predict the final assistant response $\boldsymbol{r}_n$ plus the end tag, given $\text{CONCAT}(\boldsymbol{p}_1, \boldsymbol{r}_1, \ldots, \boldsymbol{p}_n)$. Details of this process are given in Section 5.

Given $c = [\boldsymbol{p}_1, \boldsymbol{r}_1, \ldots, \boldsymbol{p}_n, \boldsymbol{r}_n]$ where $c \sim D$, we evaluate

$$\text{LLM}_{D'}(\text{CONCAT}(\boldsymbol{p}_1, \boldsymbol{r}_1, \ldots, \boldsymbol{p}_n, \boldsymbol{r}_n)).$$

If $c$ is a complete conversation lacking the end tag, in expectation the response will be end. If the prompt is an incomplete conversation where the agents fail to fully address the user's request,

in expectation the response will contain additional turns $\boldsymbol{p}_{n+1}, \boldsymbol{r}_{n+1}, \ldots$ representing the agents' remaining tasks (Figure 1). This approach has the additional benefit of summarizing remaining tasks in $\boldsymbol{p}_{n+1}, \boldsymbol{r}_{n+1} \ldots$, which can provide insights into the types of tasks the LLM fails to complete.

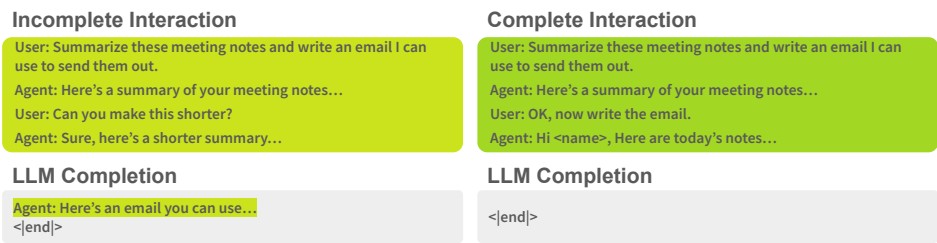

Figure 1: Identifying incomplete interactions via LLM completion of the full interaction.

## 4.3 RESPONSE TREES

Finally, we measure the LLM's uncertainty throughout the conversation. This is useful because uncertainty is intuitively related to underspecified goals, in which the human does not provide enough information for the LLM to respond correctly, and unusual or out-of-domain questions that the LLM may answer incorrectly because they are not well-represented in the training data. We construct a *response tree*: a graph that approximates the conditional probability distribution $P_D(\mathbf{r} \mid \mathbf{p} = \boldsymbol{p})$ for a given prompt $\boldsymbol{p}$.

While a complete instantiation of this distribution is prohibitively expensive to generate, given a threshold probability $\alpha$, we can generate all branches of the tree that could be traversed with probability $\geq \alpha$. We define $\text{RTREE}_{D,\alpha}(\boldsymbol{p})$ as the function that returns the response tree of prompt $\boldsymbol{p}$ for distribution $D$ and threshold $\alpha$. A detailed generation procedure is provided in Appendix B. In practice, we constrain the number of branches using $\alpha$ and limit their depth to reduce the computational complexity of generating the tree.

The response tree represents a technical improvement over evaluating the log probability of a generated response because it provides an overview of multiple possible responses without repeated high-temperature sampling. Human inspection of the response tree can also yield insights into the nature of potential errors. Figure 6 shows a simplified example of this phenomenon.

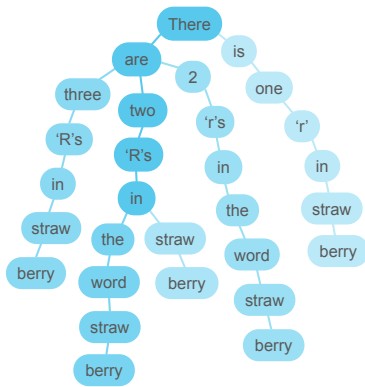

Figure 2: Simplified response tree for the prompt "How many 'r's are in the word strawberry?". Lighter branches correspond to less probable responses.

**Justification:** Suppose an LLM response tree for prompt $\boldsymbol{p}$ has one branch representing response $\boldsymbol{r}$, which will be generated with probability $\geq 1 - \alpha$. This response will be consistently generated in response to prompt $\boldsymbol{p}$ [2].

Now suppose an LLM response tree has $n$ branches representing responses $\boldsymbol{r}_1, \ldots, \boldsymbol{r}_n$, such that $\boldsymbol{r}_1, \ldots, \boldsymbol{r}_n$ make up at least $1 - \alpha$ of the probability mass for the distribution of responses to prompt $\boldsymbol{p}$. If one were to sample many responses, one would expect to see $\boldsymbol{r}_1, \ldots, \boldsymbol{r}_n$ in the resulting samples. Since $\boldsymbol{r}_i \neq \boldsymbol{r}_j$, the increased number of branches reflects uncertainty in the response to prompt $\boldsymbol{p}$. Assuming there is one correct response, a large number of leaf nodes corresponds to an increased likelihood that an incorrect response will be generated.

**Even if one does not know which response is correct, one can infer there is a high probability of error** by the presence of multiple different responses.

---

[2] It is possible for $\boldsymbol{r}$ to be an incorrect response to $\boldsymbol{p}$; this would not be detectable by our metric, and would occur if the LLM is highly confident in an incorrect answer.

## 5 Results

### 5.1 Datasets

We evaluate our metrics on datasets spanning a range of general and specific-purpose tasks. We selected LMSYS-Chat-1M (Zheng et al., 2024) to represent unstructured chats between real humans and LLMs; code-feedback (Zheng et al., 2025), a dataset used to fine-tune the OpenCodeInterpreter LLM, to represent code generation applications; and several synthetic datasets representing specific tasks: multi-turn insurance underwriting (AI, 2025c) and four tasks from the Agent-FLAN dataset (Chen et al., 2024), which standardizes the interaction format and adds a "supervisor" agent that guides the conversation. We select online shopping (Yao et al., preprint), SQL query-writing, Q&A with a knowledgebase (KB), and interacting with a computer terminal (OS) (Liu et al., 2023a). Because the SQL, OS, and KB datasets are relatively small, we combine them to represent an agent with multiple technical skills. Samples from each dataset are provided in Appendix C.

| Dataset | Size | Topic | Objective-Driven | Tool Use |
|---|---|---|---|---|
| LMSYS-Chat-1M | 1000 | Unstructured human-LLM chats | ✗ | ✗ |
| Code-Feedback | 1000 | Code generation and debugging | ✓ | ✗ |
| Insurance | 380 | Insurance underwriting | ✓ | ✓ |
| WebShop | 351 | Web interaction for shopping | ✓ | ✓ |
| SQL | 537 | Formulating SQL queries | ✓ | ✓ |
| OS | 195 | Terminal interaction | ✓ | ✓ |
| KB | 311 | Q&A with knowledgebase | ✓ | ✓ |

Table 1: Summary of datasets representing different applications. The WebShop, SQL, OS, and KB datasets are subsets of the Agent-FLAN dataset. For LMSYS and Code-Feedback, we select a random subsample; for all others, we use the full dataset.

**Data Preparation** In Code-Feedback, Insurance, WebShop and SQL+OS+KB (Table 1), almost all of the interactions are complete. We truncate interactions to produce incomplete samples; if a sample has $n$ turns, we draw a random integer in $1, \ldots, n-1$ and truncate the sample to that length.

**Fine-Tuning** We fine-tune four models from a base LLaMA3-8B-Completion model (Grattafiori et al., 2024) to approximate the token distributions of Code-Feedback, Insurance, WebShop, and SQL+OS+KB. For an $n$-turn interaction we use $\text{CONCAT}(n\boldsymbol{p}_1, \boldsymbol{r}_1, \ldots, \boldsymbol{p}_n)$ as the input and $\boldsymbol{r}_n$ as the desired output. We use supervised fine-tuning (Ouyang et al., 2022) to train a LoRA adapter on 50% of each dataset, reserving the other 50% as unseen test data. Each model is trained for 3 epochs using the AdamW 8-bit optimizer (Loshchilov & Hutter, 2017) with a learning rate of 0.0002 and weight decay of 0.01. For datasets that do not require fine-tuning, we prompt LLaMA3.1-8B-Instruct to generate completions. We selected these models to demonstrate an additional benefit of our approach: the use of **small models for LLM evaluation** where previously, larger models were required for LLM-as-a-judge techniques (Gu et al., 2024).

### 5.2 Goal Clustering

We assess the stability of our unsupervised clustering approach by running multiple trials, showing that for all datasets except Code-Feedback and Insurance, our algorithm produces highly stable clusters (Figure 4). Inspection of the cluster labels in Appendix C suggests these two datasets are difficult to label because they contain multiple topics: Insurance could be labeled by underwriting task or by the type of business being insured, and Code-Feedback could be labeled by the type of question or programming language.

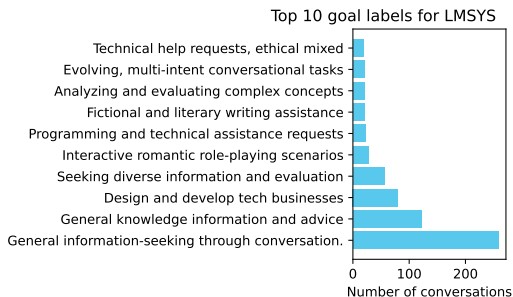

Figure 3: Top 10 largest clusters for LMSYS

We report the largest top 10 clusters in (Figures 3 and 5) and the full clustering assignments for two runs in Appendix C. LMSYS, Code-Feedback

and KB+OS+SQL exhibit many small data clusters as they contain a wide range of topics. Insurance and WebShop contain a smaller number of labels.

We compare our algorithm against a baseline LLM-only approach on a 250-sample subset of each dataset. In the baseline, we prompt OpenAI's GPT-4.1 (OpenAI et al., 2024) to label each sample with either a previously defined label or a new one (the prompts for this approach are given in Appendix C). Figure 4shows that this baseline is unstable and produces different clusters based on the order of the data. For WebShop, it produces a single cluster "Online Shopping and Purchase". Finally, to quantify the cluster stability, we report the Adjusted Rand Index (ARI) Al-jabery et al. (2020), a measure of agreement between two sets of cluster labels. ARI takes the value 1 when the two sets of labels agree perfectly, and 0 when their agreement is no better than the agreement produced by random chance. Because our clustering algorithm and the LLM judge baseline can both produce different numbers of clusters with different names, we match names using embedding similarity with `text-embedding-3-small` to identify consistent clusters across multiple runs of the same algorithm. Table 2 shows our algorithm significantly outperforms a GPT-4.1 LLM judge.

|  | LMSYS | Code-Feedback | Insurance | WebShop | KB+OS+SQL |
|---|---|---|---|---|---|
| Ours | $0.7 \pm 0.06$ | $0.33 \pm 0.06$ | $0.27 \pm 0.09$ | $1.0 \pm 0.0$ | $1.0 \pm 0.0$ |
| LLM Judge | $0.01 \pm 0.06$ | $0.03 \pm 0.10$ | $0.01 \pm 0.05$ | $1.0 \pm 1.0$ | $-0.01 \pm 0.05$ |

Table 2: ARI comparing multiple runs of our clustering algorithm, or multiple runs of an LLM-as-a-judge baseline, with confidence intervals generated by bootstrapping. On the WebShop dataset, the LLM judge produces a single cluster which trivially yields perfect agreement across multiple trials.

## 5.3 COMPLETION LABELING

To evaluate LMSYS, which represents interactions from the base chat distribution, we use Llama3-8B-Instruct with a short prompt (Appendix B). We omit this prompt for the fine-tuned models. Because LMSYS is not labeled, we select 100 samples to annotate and use as ground-truth labels for the statistics in Table 3). For Code-Feedback, Insurance, WebShop and SQL+OS+KB, we treat full samples as complete and truncated samples as incomplete.

We report accuracy, precision, recall, and F1-score of our completion labeling method and present ablation tests showing the value of fine-tuning. Or method, using an 8B fine-tuned model, matches or exceeds the performance of a 70B LLM judge (Table 3). We observe that while the other task-oriented datasets are easily classified, no completion method performs well on Code-Feedback, likely because many samples contain a programming problem answered in the first turn and follow-up questions in subsequent turns. Thus there is no well-defined end, as another follow-up is always plausible. The same problem is present in some LMSYS chats. On the Insurance dataset, our fine-tuned model produces inconsistent performance. We attribute this to the small training dataset (50% of 380 samples) and the complexity of the insurance underwriting task. For comparison, the WebShop dataset contains only 351 samples but follows a simpler format, and SQL+OS+KB contains 1043 samples.

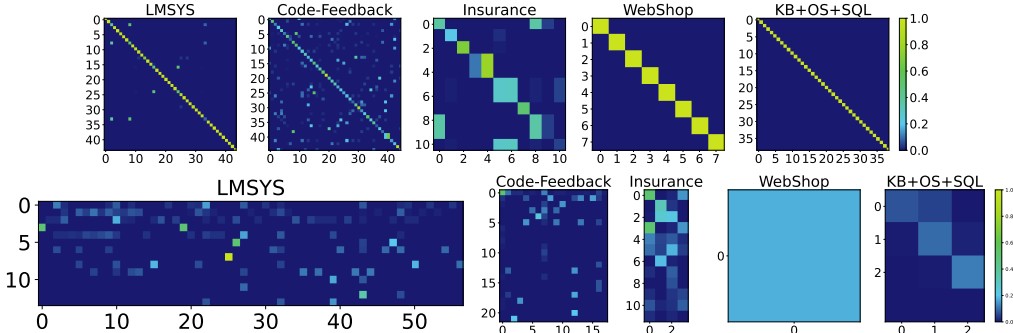

Figure 4: Labeling confusion matrices for two runs of LLM-supervised clustering (top) and an LLM-only labeling baseline (bottom). To visualize changes across two clustering runs, we compute a matrix $D$ where $D_{ij}$ is the number of elements in both cluster $i$ for run 1 and cluster $j$ for run 2, then sort the matrix to align the largest elements on the diagonal.

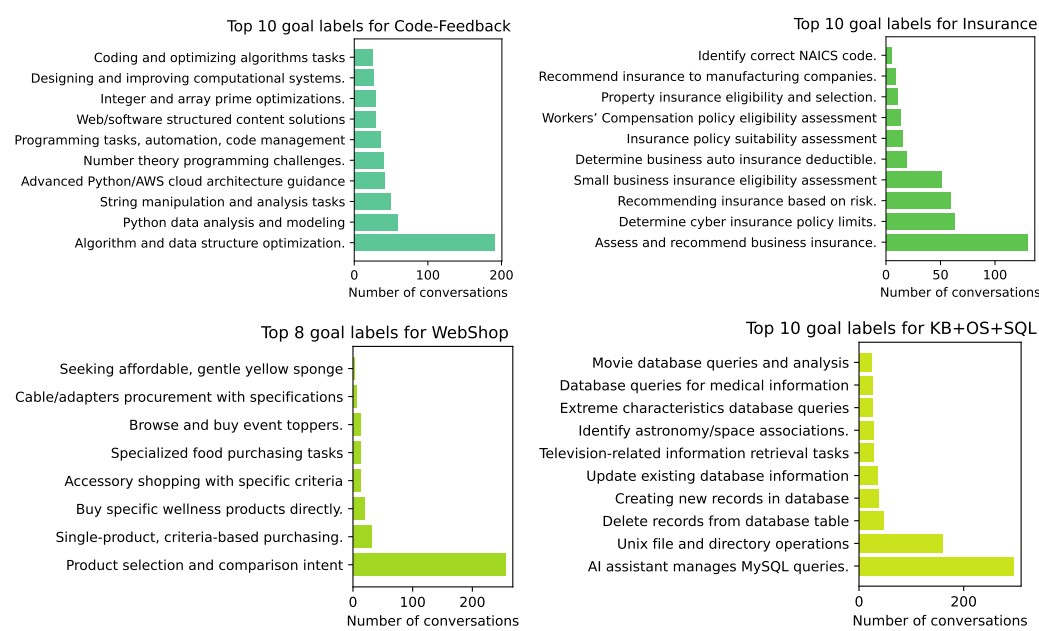

Figure 5: Top 10 largest clusters for objective-driven datasets. We report full labels in Appendix C.

Finally, the "Insurance (No end tag)" row in Table 3 represents an ablation test in which we fine-tuned the LLaMA3.2-8B completion model on the Insurance dataset without the end tag. Its poor performance shows the necessity of the end tag to mark a completed interaction.

We also report $p$-values in Table 3 for the null hypothesis that our method's accuracy is no better than the 70B LLM judge, showing a statistically significant difference on all datasets except SQL+OS+KB, where both methods are highly accurate[3]

One natural follow-up question is whether we can relate the performance of this approach to the percentage of incomplete conversations in the training dataset. However, to conduct this analysis one would need to relate the prevalence of certain tokens in a dataset to the probability of generating said tokens after fine-tuning on the dataset. Unfortunately, theoretical analysis of fine-tuning remains an understudied question; thus analysis of this approach would require extensive additional work.

## 5.4 RESPONSE TREES

We construct response trees for our datasets using the same base or fine-tuned models from Section 5.3. We prompt the models with $\text{CONCAT}(\boldsymbol{p}_1, \boldsymbol{r}_1, \ldots, \boldsymbol{p}_n)$ and construct a response tree for $\boldsymbol{r}_n$. For each sample, we measure the number of leaf nodes in the tree and the log-probability (logprob) of the highest-probability response.

While there are no ground-truth labels for the presence of errors, LLM uncertainty, or sample dissimilarity from the dataset's token distribution, we emphasize that the response tree and its statistics **quantify the empirical distribution** of $P(\mathbf{r} \mid \mathbf{p} = \boldsymbol{p})$. We present aggregate statistics on our datasets in Figure 6, which show that despite fine-tuning, our models exhibit high uncertainty on the KB+OS+SQL and Insurance datasets, with the KB+OS+SQL dataset being most challenging. This dataset is highly dissimilar from the base chat distribution due to the prevalence of tool calls, SQL, terminal interactions and tool output. Our models show higher confidence on LMSYS and Code-Feedback, reflected by a high max logprob value. Finally, Table 4 shows that the max logprob

---

[3]Because it would be both computationally intensive to redo the fine-tuning step for many cross-validation splits, and an invalid comparison as the LLM judge has no training step and the size of the fine-tuning dataset affects the resulting model's performance, we use one fine-tuned model and evaluate the accuracy of random subsamples of each dataset to produce the reported $p$-value via a two-sided t-test.

| Dataset (Evaluator) | Accuracy | Precision | Recall | F1 | $p$-val |
|---|---|---|---|---|---|
| LMSYS subsample (LLama 70B judge) | 0.43 | 0.77 | 0.25 | 0.38 | |
| LMSYS subsample (LLama 8B completion) | **0.74** | **0.79** | **0.85** | **0.82** | 0.005 |
| Code-Feedback (LLaMA 70B judge) | 0.53 | 0.53 | 0.46 | 0.49 | |
| Code-Feedback (LLaMA 8B completion) | **0.59** | 0.56 | **0.84** | **0.67** | |
| Code-Feedback (Fine-tuned 8B completion) | 0.47 | **0.71** | 0.12 | 0.21 | 0.074 |
| Insurance (LLaMA 70B judge) | **0.95** | **1.0** | **0.91** | **0.95** | |
| Insurance (LLaMA 8B completion) | 0.80 | 1.0 | 0.60 | 0.75 | |
| Insurance (No end tag) | 0.70 | 0.66 | 0.79 | 0.72 | |
| Insurance (Fine-tuned 8B completion) | 0.91 | 0.94 | 0.87 | 0.91 | $1.4e^{-5}$ |
| WebShop (LLaMA 70B judge) | **0.92** | **1.0** | 0.83 | 0.91 | |
| WebShop (LLaMA 8B completion) | 0.66 | 0.64 | 0.71 | 0.68 | |
| WebShop (Fine-tuned 8B completion) | **0.92** | 0.89 | **1.0** | **0.94** | $5.6e^{-5}$ |
| SQL + OS + KB (LLaMA 70B Judge) | 0.97 | 0.96 | 0.97 | 0.96 | |
| SQL + OS + KB (LLaMA 8B completion) | 0.52 | 0.51 | 0.92 | 0.66 | |
| SQL + OS + KB (Fine-tuned 8B completion) | **0.98** | **0.99** | **0.98** | **0.99** | 0.44 |

Table 3: Comparison of our completion labeling method against several baselines, showing that our method can outperform a 70B LLM with an 8B fine-tuned model and generalizes to unstructured, non-objective driven chats. We report the $p$-value for the null hypothesis that our method performs no better than the 70B LLM judge.

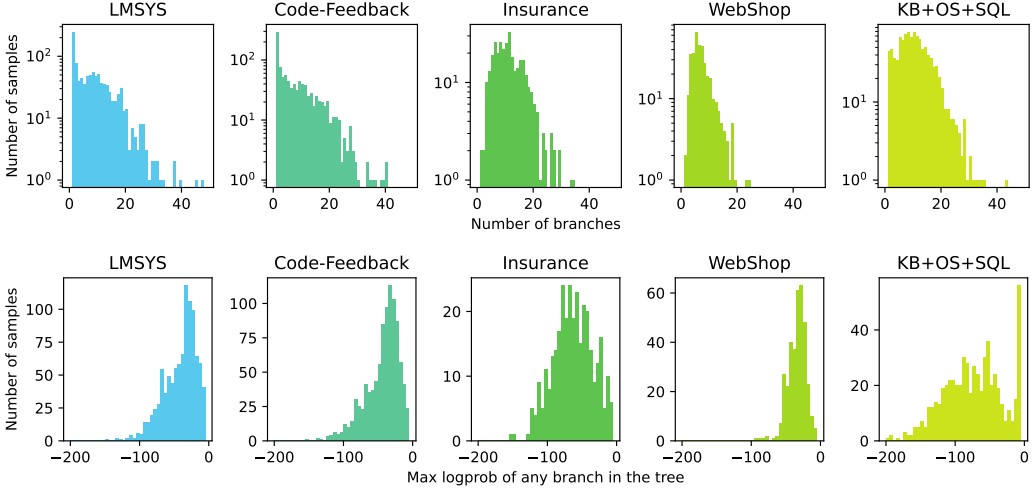

Figure 6: Histograms of the number of response tree branches for each sample (top) and the max logprob for each sample, defined as the max logprob for any branch in RTREE(CONCAT($\boldsymbol{p}_1, \boldsymbol{r}_1, \ldots, \boldsymbol{p}_n$)).

and number of leaves are not correlated with the interaction length, validating that the response tree is extracting more complex information about LLM uncertainty.

| | LMSYS | Code-Feedback | Insurance | WebShop | KB+OS+SQL |
|---|---|---|---|---|---|
| Max logprob vs length | -0.11 | -0.19 | -0.25 | 0.16 | 0.41 |
| Max logprob vs leaf nodes | -0.49 | -0.46 | -0.10 | -0.19 | -0.06 |

Table 4: Correlations between maximum logprobs, number of leaf nodes, and conversation lengths.

## 6 LIMITATIONS

Our provided clustering approach relies on two inputs: the prompt used to summarize interactions, which can be modified to focus on desired attributes of the interaction, and the initial number of clusters $k_1$, which limits the maximum number of clusters that can be discovered. Currently, our

approach does not perform multiclass classification but could be extended to do so, which would likely yield better results on datasets such as Code-Feedback and Insurance. Additionally, as shown in Table 3, our method of completion labeling relies on structure in the interaction; a common property of objective-driven interactions. Thus, it does not perform well on datasets where the user's goal is completed in the first turn and subsequent turns contain follow-up questions.

# 7 IMPACT

We present novel, statistically grounded metrics for evaluating complex enterprise AI systems, a critical need as these systems are already in use in many industries. We show via ablation studies that our metrics can outperform LLM-as-a-judge evaluation using only LLaMA3-8B models. Beyond this immediate application, the use of small LLMs means our metrics lend themselves to future work developing online monitoring and intervention systems. Measuring uncertainty can provide a signal to ask a human for assistance, avoiding errors and wasted tokens. Labeling completion can also be used to save computing resources: a small LLM could monitor the output of a larger LLM, detect when the interaction is likely to be complete, then prompt the larger LLM to finish. Our work provides a foundation for these online interventions.

# 8 FUTURE WORK

One novel contribution we make is the use of fine-tuned LLMs to model specific token distributions. A future line of work concerns the statistical properties of this approach; for example, how the distance between the "base" distribution (the distribution of text the LLM is pretrained on) and the fine-tuning distribution affects performance; how the fine-tuning dataset size affects the model's performance on completion labeling; and whether one can establish statistical guarantees. We also introduce the response tree as an approximation of $P(\mathbf{r} \mid \mathbf{p} = \boldsymbol{p})$. While we use the response tree to quantify uncertainty in LLM responses, we expect it to prove useful in other applications; for example, if one knows the sampling strategy used by an LLM, one can establish a statistical guarantee on the probability of an LLM producing a certain output by inspecting the response tree. One could even modify this probability by developing a training strategy that takes into account the entire tree instead of one sampled branch. We are excited to explore these ideas in future research.

# 9 CONCLUSION

The novel contribution of this work is a suite of three weakly supervised metrics for evaluating multi-turn objective-driven conversations, a task of critical importance for enterprise AI developers. Because objective-driven interaction data from these systems is not publicly available, we validate our method on a combination of real chat data and synthetic objective-driven multi-turn data, including distribution shifts representing tool use and specialized agents for business tasks. We provide a novel LLM-guided clustering algorithm for text data, which combines k-means with LLMs to produce stable, interpretable clusters. Additionally, we introduce the LLM as a model of a token sequence distribution: a framework that enables weakly supervised evaluation in challenging settings with small, efficient models fine-tuned to adapt to distributional shifts.

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

## A  BACKGROUND

### A.1  LLMs AS APPROXIMATORS OF A TOKEN SEQUENCE DISTRIBUTION

We use the following notation. A *token* is a discrete value drawn from a finite *vocabulary* $\mathbb{L}$. A prompt is an ordered list of tokens $\boldsymbol{p} = p_1, \ldots, p_N$, where $N$ is the *context length*, and a response is an ordered list of tokens $\boldsymbol{r} = r_1, \ldots, r_M$, where $M$ is the *maximum output length*. Unused elements in these sequences are filled by a "pad" token, mirroring the actual implementation of open-source LLMs. We use $\mathbf{P}$ and $\mathbf{R}$ to denote the random variables corresponding to the prompt and response, and $P_W(\mathbf{R} \mid \mathbf{P})$ to denote the conditional probability distribution of $\mathbf{R}$ given $\mathbf{P}$ under distribution $W$ (the "world" of possible prompt-response pairs). This definition emphasizes the probabilistic nature of the LLM response.

LLMs use a variety of sampling strategies to select a single response $r \in \mathbb{L}^M$ from $P_W(\mathbf{R} \mid \mathbf{P} = \boldsymbol{p})$ (the conditional probability distribution for prompt $\boldsymbol{p} \in \mathbb{L}^N$. For our analysis we assume the LLM selects the highest probability response, though in practice this strategy is modified to prevent malformed responses, such as repetitions of high-probability words, from being generated. We define $\text{LLM}_W : \mathbb{L}^N \to \mathbb{L}^M$ as the function that takes as input a prompt $\boldsymbol{p}$ and returns the most probable response $\boldsymbol{r}$.

### A.2  EXTENSION TO MULTI-TURN CONVERSATIONS

Thus far, we have considered single-turn conversations consisting of one user prompt $\boldsymbol{p}$ and one LLM response $\boldsymbol{r}$. A multi-turn conversation consists of an ordered list of sequences $\boldsymbol{p}_1, \boldsymbol{r}_1, \ldots, \boldsymbol{p}_T, \boldsymbol{r}_T$. To extend our definition of $\text{LLM}_W$ to a multi-turn setting, we define $\text{CONCAT}(\boldsymbol{p}_1, \boldsymbol{r}_1, \ldots, \boldsymbol{p}_k)$ as the concatenated sequence of $\boldsymbol{p}_1, \boldsymbol{r}_1, \ldots, \boldsymbol{p}_k$ and:

$$\boldsymbol{r}_k = \text{LLM}_W \left( \text{CONCAT}(\boldsymbol{p}_1, \boldsymbol{r}_1, \ldots, \boldsymbol{p}_k) \right).$$

## B  IMPLEMENTATION DETAILS

### B.1  UNSUPERVISED CLUSTERING

We provide examples of the intent summaries used for clustering in Algorithm 1. Where application context paragraphs are used, we provide those as well. We do not use application context paragraphs for LMSYS or Code-Feedback due to their similarity to the base chat distribution.

**LMSYS Goal summary; no application context**

```
INTENT = "Based on the input log, the user's high-level intent can be
described as follows:\n\n1. The user is seeking to understand the
concept of stepwise refinement in a program development process and is
looking for the correct approach that involves breaking down the program
into stages.\n2. The user is trying to identify a specific time-related
measure in manufacturing, specifically the time it takes to complete one
cycle of production from start to finish."
```

**Code-Feedback Goal summary; application context**

```
INTENT = "Based on the input log, the user's high-level intent can be
described in the following sentences:\n\n1. The user wants to remove
duplicates from an array while maintaining constant space complexity,
meaning no additional data structures should be used.\n2. The user
initially wants to achieve this using a two-pointer technique, but
later realizes that the approach is not efficient for large lists due
to its quadratic time complexity.\n3. The user then asks for an
optimized solution that adheres to constant space complexity and has
a better time complexity.\n4. The user is open to using a set or
dictionary for the task, but the assistant suggests an alternative
```

```
11   method that uses sorting and in-place operations to achieve the
12   desired result."
```

**Insurance Goal summary and application context**

```
1   CONTEXT = "You will see a chat between a human insurance underwriter and
2   an AI assistant. The assistant's task is to assist the human in
3   determining whether to offer a customer an insurance policy, and if so,
4   what type of policy to offer. The assistant may use tools to retrieve
5   information."
6   INTENT = "The user's **high-level intent** is to determine whether a
7   specific company qualifies as a small business in order to offer them
8   a workers' compensation insurance policy."
```

**WebShop Goal summary and application context**

```
1   CONTEXT = "You will see a chat between a user and an AI shopping
2   assistant. The assistant can search the web for products, read
3   information from product pages and click on links to navigate
4   shopping websites and make purchases at the user's request."
5   INTENT = "The user's **high-level intent** is to purchase a three-
6   piece, wall-mounted, stainless steel spice rack for less than $30.00."
```

**KB+OS+SQL Goal summary and application context**

```
1   CONTEXT = "You will see a chat between a user and an AI assistant. The
2   assistant has three capabilities: it can interact with a computer
3   terminal and read the output, it can formulate and run SQL queries,
4   receiving the results as text, and it can query a knowledgebase to
5   receive information that will assist it in answering the user's
6   questions."
7   INTENT = "The user's high-level intent is to retrieve a list of movie
8   IDs acted by the top 5 actors based on their ratings."
```

**Phase 1 Clustering Prompt Example - Insurance**   We show three samples for readability, however in practice we show ten samples for the group and ten negative examples.

```
1    You will see a list [GROUP] of task descriptions that all have something
2    in common, and a list [NOT IN GROUP] of task descriptions that do not
3    fit with the group. Please summarize the [GROUP] in one sentence. The
4    sentence should describe tasks in the [GROUP], and should not describe
5    tasks listed under [NOT IN GROUP].
6    [GROUP]
7    - Based on the input log, the user's high-level intent can be described
8    in two concise sentences:
9    1. The user wants to determine which lines of business (LOBs) make sense
10   for a company that produces and sells compost, specifically Green Earth
11   Compost.
12   2. The user needs to assess whether Green Earth Compost qualifies as a
13   small business and meets the underwriting guidelines for various LOBs to
14   provide comprehensive coverage for their operations.
15
16   - Based on the input log, the user's high-level intent can be described
17   in two concise sentences:
18   1. The user is seeking to identify potential additional lines of business
19   (LOBs) that the company, Green Earth Excavating, might be interested in
20   purchasing.
21   2. The user is looking for guidance on which LOBs are already offered or
22   bound for Green Earth Excavating, given their industry and current LOB.
23
```

```
- Based on the input log, the user's high-level intent can be described
in two concise sentences:
1. The user wants to determine if a specific customer is in appetite for
the chosen LOB (Cyber).
2. The user needs to verify if the customer qualifies as a small
business, as the guidelines state that only small businesses are in
appetite.

[NOT IN GROUP]
- The user's **high-level intent** is to determine if the insurance
company will cover the magazine publisher's General Liability policy.

- The user's **high-level intent** is to obtain a recommendation for
policy limits for a cyber liability insurance policy for a small
business, specifically an optometry office with retail eyewear sales.

- The user's **high-level intent** is to determine the correct policy
limits for a company's Cyber Line of Business (LOB) based on its
specific characteristics and underwriting guidelines.
```

**Phase 2 Cluster Merge Prompt Example (clusters were merged)**   We show three samples for readability, however in practice we show ten samples for the group and ten negative examples.

```
You will see two groups of task descriptions, and a third list of
unrelated tasks that aren't in either group. Your goal is to determine
whether the two groups should be merged. If so, determine a description
for the merged groups that accurately describes them and does not
describe the unrelated tasks.
[GROUP 1]
- The user's **high-level intent** is to determine which lines of
business (LOBs) make sense for a short-term inventory finance company,
based on its characteristics and underwriting guidelines.

- The user's **high-level intent** is to identify additional lines of
business (LOBs) to present to a client, specifically a wholesale
distributor, in order to provide a comprehensive insurance coverage.

- The user's **high-level intent** is to determine which lines of
business (LOBs) are available for a company that provides short-term
inventory finance and trade lending, and to understand the appetite
for each LOB based on the company's characteristics.

[GROUP 2]
- This group is empty - merge it.

[NOT IN GROUP]
- The user's **high-level intent** is to determine the standard limit
for a
Cyber insurance policy for a small outpatient healthcare company with
$6.7 million in annual revenue.

- The user's **high-level intent** is to determine the correct six-digit
NAICS (North American Industry Classification System) code for a mortgage
lender specializing in home equity lines of credit.

- The user's **high-level intent** is to determine if the insurance
company is interested in covering a specific line of business (LOB) for
a particular business, Midwest Insight Research LLC.

Additionally, the user's intent is to gather information about the
business, such as its NAICS code, annual revenue, and location, in
order to make this determination.
```

## B.2 Completion Labeling

We use the following prompt to evaluate completion using a non-fine-tuned instruction-following LLM (LLaMA3.2-8B-Instruct). `USER CHAT: Concisely summarize the remaining tasks. If there are no more tasks, output <end of system logs>. ASSISTANT CHAT: .`

In our experiments, the fine-tuned models output an incorrect (but consistent) end tag. We present some examples of completed and incomplete end tags. We note that the fine-tuned models often produce ungrammatical outputs, which we attribute to the use of LLaMA3-8B models (imperfect, but efficient approximators of the token sequence distribution).

**LMSYS Completion examples (not fine-tuned)**

```
1  "completion": "TURN 7, STEP 13, assistant chat:\nThe remaining tasks are:
2  - Provide an interactive lesson to learn English about the development of
3  AI (already provided) (complete, incorrectly labeled as incomplete)
4  completion": "<end of system logs> (complete)
5  "completion": "TURN 4, STEP 7, user chat:\nGenerate a smutty scenario for
6  me. (incomplete)
```

**Code-Feedback Completion examples**

```
1  "completion": "end system\n", (complete)
2  "completion": " TURN4 STEP7 user:\n (complete, incorrectly labeled
3  ncomplete)
4  {"completion": "TURN2 STEP3 , chatThe has successfully the code.
5  (truncated)
6  "completion": "java\njava\njavajavajavajava (truncated)
```

**Insurance Completion examples**

```
1  "completion": "." (complete)
2  "completion": "., you help determine appetite these products (complete,
3  incorrectly labeled incomplete)
4  "completion": "., could tell if General is appetite for applicant.
5  (truncated)
6  "completion": ", I help that. is business as small?, (truncated)
```

**WebShop Completion examples**

```
1  "completion": "end system" (complete)
2  "completion": "Tr for  barst  height faux \n (truncated)
3  "completion": "Tr ,  steps  to  in chat  with  and  of  lights  glass
4  price $.00 (truncated)
5  "completion": "end system\n\n  [](https://i.stack.imgur.com/Q280X.png)
6  (truncated, incorrectly labeled complete)
```

**KB+OS+SQL Completion examples**

```
1  "completion": "end system" (complete)
2  "completion": "sTURN4 STEP7 user: (truncated)
3  "completion": "TURN3 STEP5 user:\n (truncated)
4  "completion": "s TURN5 STEP9 user:\nle1 which instances basetools_typ
5  (truncated)
```

### B.3 Response Tree Generation with LLaMA Models

The LLaMA family of models uses a chat *template*: a set of special tokens that demarcate user identities annd the start and end of text input Meta (2025). Concretely, LLaMA models are trained on inputs and outputs that follow the chat template, and perform best when inputs follow the same template.

Listing 1: LLaMA 3 Chat Template

```
1  <|begin_of_text|><|start_header_id|>system<|end_header_id|>
2
3  Cutting Knowledge Date: December 2023
4  Today Date: 23 July 2024
5
6  You are a helpful assistant<|eot_id|><|start_header_id|>user<|
       end_header_id|>
7
8  How many 'r's in strawberry?<|eot_id|><|start_header_id|>assistant<|
       end_header_id|>
9
10 There are two 'r's in the word strawberry.<|eot_id|>
```

To generate a response tree, we must modify this template to correctly display a partial assistant response for completion.

Listing 2: Generating a partial assistant response for "How many 'r's in strawberry?" with the leading tokens "There are"

```
1  <|begin_of_text|><|start_header_id|>system<|end_header_id|>
2
3  Cutting Knowledge Date: December 2023
4  Today Date: 23 July 2024
5
6  You are a helpful assistant<|eot_id|><|start_header_id|>user<|
       end_header_id|>
7
8  How many 'r's in strawberry?<|eot_id|><|start_header_id|>assistant<|
       end_header_id|>
9
10 There are
```

Having established this capability, generating the response tree requires generating one response and its top $k$ logprobs, then generating branching responses for any token where any of logprobs $2, \ldots, k$ fall above the branching probability threshold $\alpha$. We repeat this process recursively until no more logprobs fall above $\alpha$ or a compute threshold is reached.

## C Additional Results

### C.1 Unsupervised Clustering

We assess the suitability of the cluster for classifying new data by labeling 50% of each dataset and assigning the other 50% of the samples to existing clusters. We report the average distance of each unseen sample to its cluster center in embedding space, showing that new samples are close to existing cluster centers in embedding space (Table 5).

The distances from cluster centers reported in Table 5 yield insight into the nature of the data: the Code-Feedback and Insurance datasets, which are narrowly focused on their respective topics, yield tighter clusters than LMSYS, WebShop (due to the wide range of products mentioned in shopping tasks), and SQL+OS+KB (again due to the wide range of questions).

**Comparison to LLM-as-a-judge Baseline** We compare our LLM-supervised clustering algorithm to an LLM-only clustering approach where GPT-4.1 (OpenAI et al., 2024) is prompted directly to identify the user's goal.

| Dataset | # clusters (train) | Distance (train) | Distance (test) | # clusters (full) | Distance (full) |
|---|---|---|---|---|---|
| LMSYS-Chat-1M | 21 | 0.95 | 0.70 | 41 | 1.03 |
| Code-Feedback | 20 | 0.75 | 0.65 | 43 | 0.79 |
| Insurance | 6 | 0.78 | 0.51 | 11 | 0.89 |
| WebShop | 4 | 1.06 | 0.60 | 8 | 1.49 |
| SQL + OS + KB | 18 | 1.36 | 0.87 | 42 | 0.91 |

Table 5: Cluster statistics for our unsupervised clustering algorithm. To report the average train and test distances, we use a 50-50 random train-test split.

For the first sample, we use the prompt

```
Based on the [USER INPUT LOG], please describe the user's **high-level
intent** in three to five words.
[USER INPUT LOG]
  ...
```

When one or more labels have been identified, we prompt the LLM to reuse previously applied labels. We provide an example from the Insurance dataset.

```
Based on the [USER INPUT LOG], please select one or more of the
following categories to describe the user's **high-level intent**. If
the user expresses an intent that does not fit any category, you may
define a new category in three to five words.
 - Define Correct NAICS Code for Underwriting, Recommend Appropriate
 Policy Limits, Determine Small Business Eligibility
 - Determine Small Business Eligibility
 - Define Correct NAICS Code for Underwriting, Recommend Appropriate
 Policy Limits
 - Identify Additional Insurance LOB Opportunities
 - Recommend Appropriate Policy Limits
 [USER INPUT LOG]
  ...
```

**LLM-Supervised Clustering; Full Results**   We also present full clustering results for two runs over each dataset, showing that our algorithm often discovers stable clusters from unlabeled data.

## C.2   RESPONSE TREE

### C.2.1   CORRELATION WITH CONVERSATION LENGTH

We measure two salient aspects of the response tree: the logprob of the most probable response, and the number of leaf nodes in the tree. We plot these values against each other (Figure 12) and against the interaction length, measured in text characters (Figure 13). In the main paper (Table 4), we also report the measured correlations of these values, showing extremely low correlation. Thus we can conclude that the number of leaf nodes in the response tree, and the logprob of the most likely response, are not functions of the conversation length.

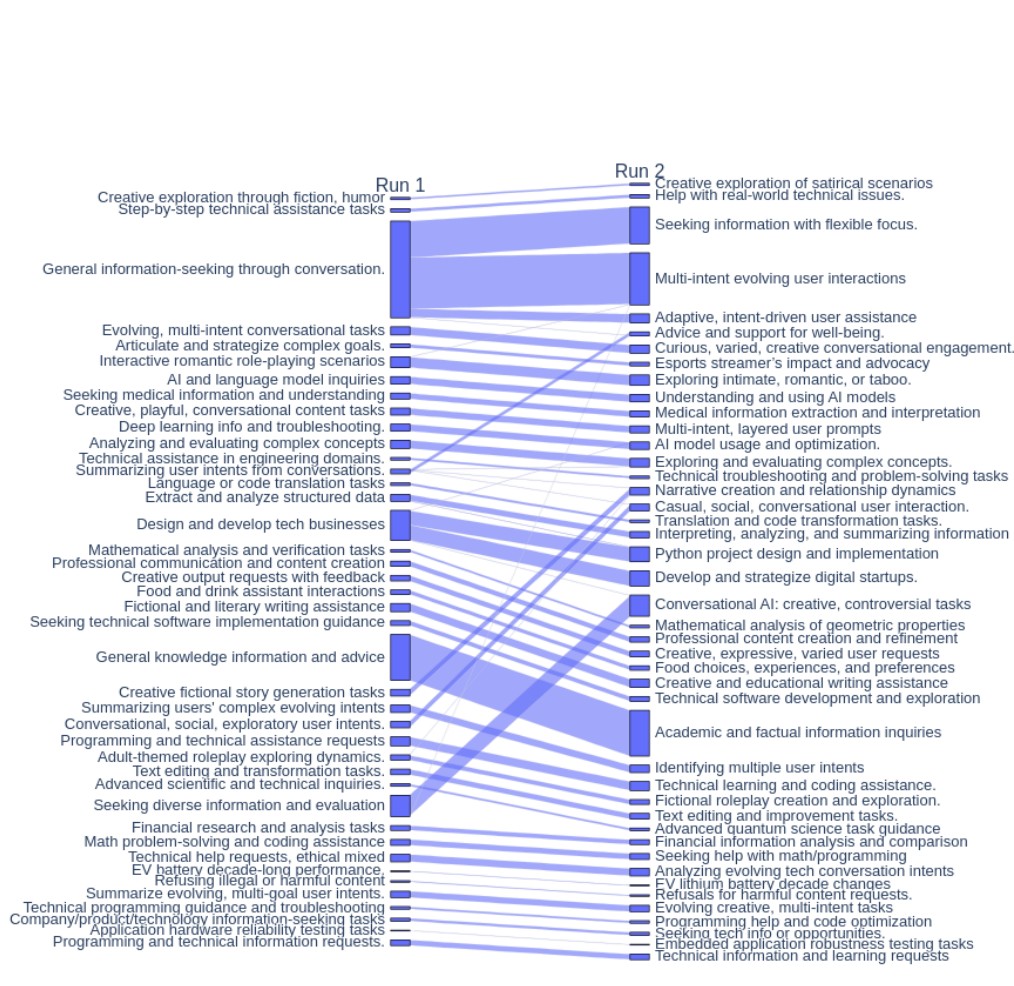

Figure 7: LMSYS label stability for two runs.

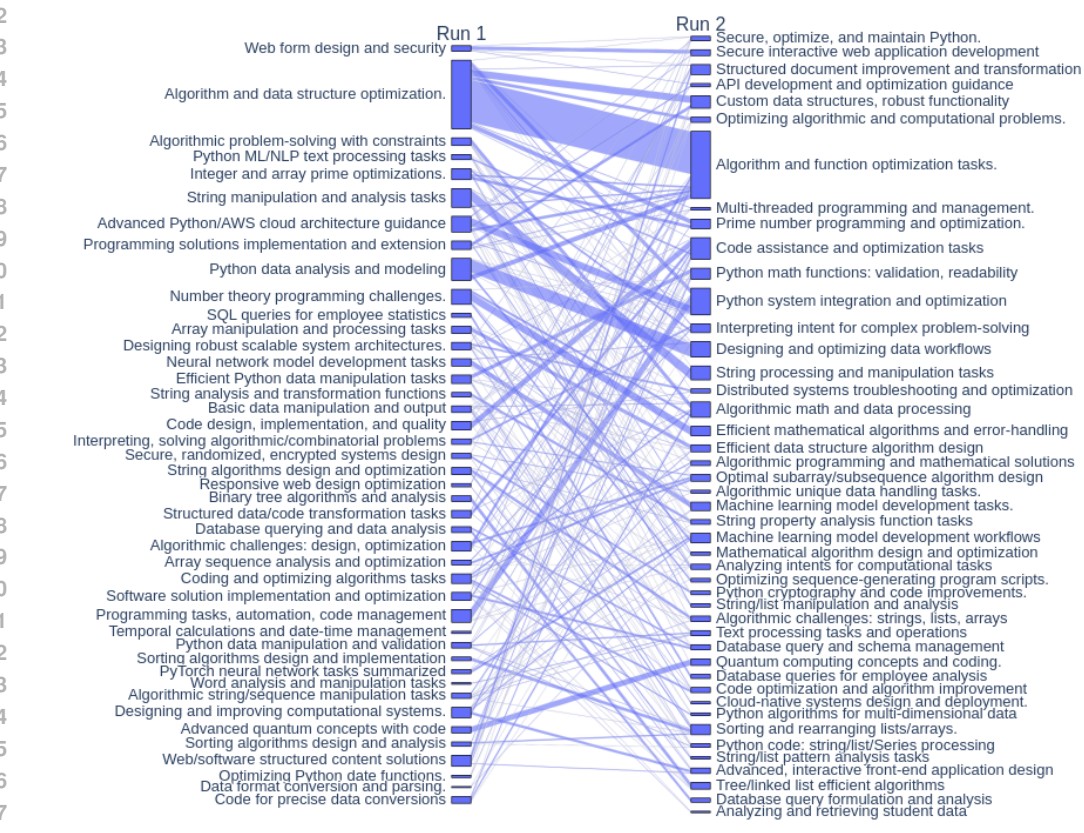

Figure 8: Code-Feedback label stability for two runs.

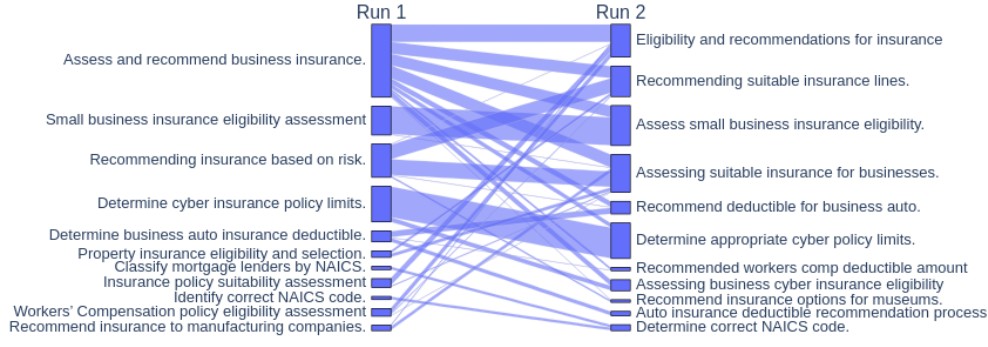

Figure 9: Insurance label stability for two runs.

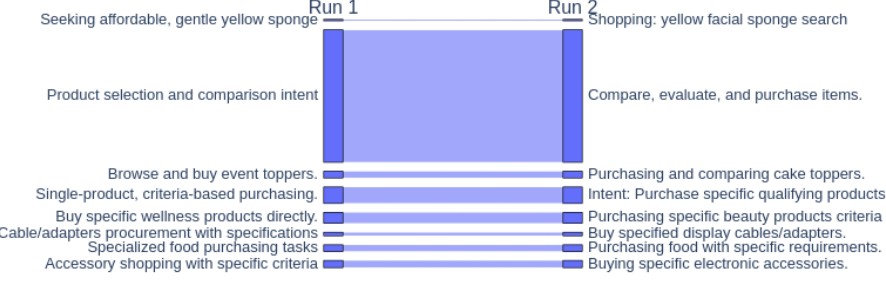

Figure 10: WebShop label stability for two runs.

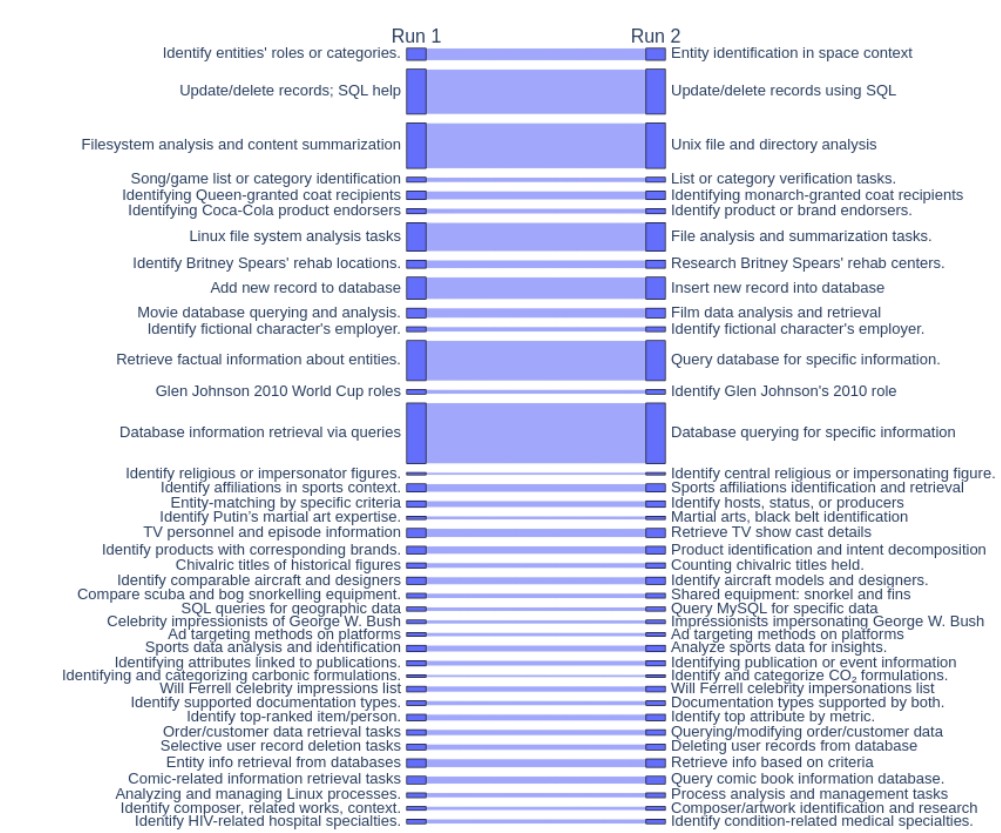

Figure 11: KB+OS+SQL label stability for two runs.

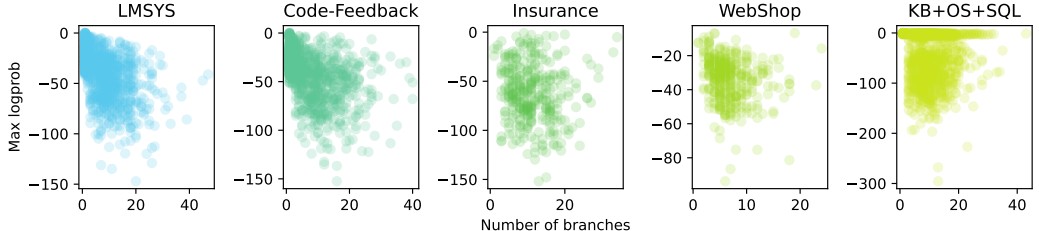

Figure 12: Max logprob vs number of leaf nodes.

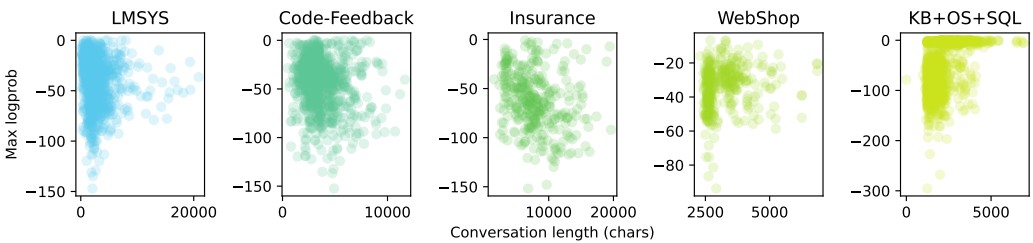

Figure 13: Max logprob vs conversation length.

### C.2.2 SAMPLES - TOP 10% OF RESPONSE TREE BRANCHES & LOGPROBS

We selected these samples by filtering for all samples that are in the 90th percentile for the number of leaf nodes in the response tree, and the 90th percentile for the negative max logprob value; e.g. these are samples where the most probable response has *low* probability and there are a *high* number of possible responses.

**LMSYS-Chat-1M, 18 branches, max logprob -12.88, completed=True, goal='General knowledge information and advice'**

```
TURN 1, STEP 1, user chat:
In which of the following,program development process is a stepwise
refinement process of the specification?
Select one:
a. Formal Specification
b. Incremental development
c. Structured programming
d. Static verification

TURN 1, STEP 2, assistant chat:
The correct answer is:
b. Incremental development.

In incremental development, the program development process is a stepwise
refinement process of the specification. It involves developing the
program in stages, with each stage building on the previous one. The
process starts with a rough outline of the program and gradually adds
more detail and functionality as the program evolves. This approach
allows for feedback and changes to be incorporated into the program as it
is being developed, making it more flexible and adaptable to changing
requirements.

TURN 2, STEP 3, user chat:
Time need from the manufacturing to turn inventory into products is known
as
Select one:
a. Cycle Time
b. Changeover Time
c. Defect Rate
d. Inventory Turnover

TURN 2, STEP 4, assistant chat:
The correct answer is:

a. Cycle Time.

Cycle time is the time it takes to complete one cycle of production,
from start to finish, including the time required to transform raw
materials into finished goods and then sell those goods to customers.
It is an important measure of efficiency in manufacturing and can
help to identify bottlenecks and areas for improvement in the
production process. Inventory turnover is a measure of how many times
he entire inventory of a company is sold and replaced over a given
period of time, and is calculated as the ratio of cost of goods sold
to average inventory.

MOST PROBABLE BRANCH:
TURN 3, STEP 5, user chat:\nWhat is the term for the process of which
```

**Code-Feedback, 53 branches, max logprob -6.46, completed=False, goal='Responsive web design optimization'**

```
TURN 1, STEP 1, user chat:
Write a CSS code to style a table such that each row alternates between
three different colors. Additionally, the first row should have a
different font color, background color, and font size than the rest of
the rows. Your CSS code should also include a hover effect on the table
rows, where the background color changes to a lighter shade when the user
hovers over a row.

TURN 1, STEP 2, assistant chat:
To achieve the desired styling for the table, you can use CSS selectors
and properties. Here's an example of CSS code that meets your
requirements:

```css
/* Define variables for the colors */
:root {
  --color-1: #f2f2f2;   /* light gray */
  --color-2: #e6e6e6;   /* gray */
  --color-3: #d9d9d9;   /* dark gray */
  --font-color-1: #333;  /* black */
}

/* Apply styles to the table */
table {

...Truncated for length...
r(--color-3);
}

/* Apply hover effect on the rows */
table tr:hover {
  background-color: #f9f9f9; /* lightest gray */
}
```

Please note that the above CSS code assumes that the table is defined in
HTML with the '<table>' element. You can apply this CSS code to your HTML
file by including it in a '<style>' tag within the '<head>' section or by
linking an external CSS file using the '<link>' tag.

TURN 2, STEP 3, user chat:
The code is well-formatted and functional, but it could be optimized.
Consider using a CSS preprocessor like SASS for more efficient handling
of color variables and nested selectors.

TURN 2, STEP 4, assistant chat:
You're right, using a CSS preprocessor like SASS can help optimize the
code by using features unavailable in regular CSS such as variables and
nested selectors. Here's how the same code would look like in SASS:

```scss
$color-1: #f2f2f2; // light gray
$color-2: #e6e6e6; // gray
$color-3: #d9d9d9; // dark gray
$font-color-1: #333; // black
$hover-color: lighten(#f2f2f2, 10%); // lighter gray for h
...Truncated for length...
child', '&:hover', '&:first-child') make it clear they're associated
with 'tr' within 'table'. Also, once the color variables at the top are
set, they can be reused throughout the stylesheet, which makes it easier
to update colors. Meanwhile, the lighten function is used to create a
lighter shade for hover effect. Please remember that SASS needs to be
compiled to CSS before serving to the browser.
```

```
TURN 3, STEP 5, user chat:
Consider optimizing the hover state to avoid unnecessary color
recalculations. Also, adding a transition effect could enhance user
interaction.

TURN 3, STEP 6, assistant chat:
Great suggestions! Here's the optimized SASS code considering your
enhancements:

```scss
$color-1: #f2f2f2; // light gray
$color-2: #e6e6e6; // gray
$color-3: #d9d9d9; // dark gray
$font-color-1: #333; // black
$hover-color: rgba(255,255,255, 0.1); // semi-transparent white to
lighten on hover

table {
  width: 100%;
  border-collapse: collapse;

  tr {
    font-size: 16px;
    color: $font-color
...Truncated for length...
er in color when hovered upon, instead of applying an entirely different
color. This approach should improve rendering performance as a separate
color calculation is no longer required for the hover state. Also, a
transition effect has been added to the background color to smoothly
animate color changes when a user interacts with a row. Again, the SASS
code needs to be compiled to CSS for web use.

MOST PROBABLE BRANCH:
lt code.<|eot_id|>
```

**Insurance, 34 branches, breaklines, max logprob = -5.05, completed = false, goal = 'Determine cyber insurance policy limits.'**

```
TURN 1, STEP 1, system chat:
You are an expert insurance underwriter. Given the following description,
assist the user with their question, taking into account your knowledge
of insurance underwriting.
Company Name: Cascadia Wellness Outpatient Center
Company Description: Cascadia Wellness Outpatient Center, located in
Everett, Washington, is a community-focused outpatient care facility
offering services such as sleep disorde
...Truncated for length...
and patient education programs to manage long-term illnesses. The center
is well-known for its comprehensive care approach and collaborative
partnerships with local health systems.
Annual Revenue: 6700000
Number of Employees: 29
Total Payroll: 1820000
Number of Vehicles: 1
Building construction: Non-combustible { Examples included steel columns,
beams and deck/floors.
State: Washington
LOB: cyber

TURN 2, STEP 2, user (underwriter) chat:
Could you find out what limit is standard for this LOB for a company
like this?
```

```
TURN 2, STEP 3, assistant (user-facing assistant) chat:
I'll need a bit more detail to pull the right guideline.

1. Which Line of Business (LOB) are you referring to?
2. What kind of company is this|e.g. its NAICS code or industry class,
size (annual revenue), or any other classification you have on hand?

TURN 3, STEP 4, user (underwriter) chat:
LOB is cyber. Size: $6,700,000 revenue.

TURN 3, STEP 5, assistant (internal assistant) action:
TOOL CALLS: [{'name': 'get_underwriting_guidelines', 'args': {}, 'id':
'call_x5DXKvCnQvZZqm9brXbUnQZu', 'type': 'tool_call'}]

TURN 3, STEP 6, assistant (tool) chat:
***General***
We write policies for small businesses only. Any business that fails to
qualify as a "small business" is out of appetite, so we will not write a
policy for it.

***Property LOB***

For property LOB, business classes pertaining to hospitality or leasing
are only in appetite if the building construction is sufficiently fire
resistive.
This means the building construction codes are eit
...Truncated for length...
te according to whether the applicant has over 20 vehicles, in which case
we decline because the exposure is too high.

***Worker's Compensation LOB***

If our appetite guide for a worker's compensation policy indicates
"qualified", then we determine final appetite according to whether the
applicant has a total annual payroll of over $5M, in which case we
decline because the exposure is too high.

TURN 3, STEP 7, assistant (user-facing assistant) chat:
To determine the appropriate cyber limits, I need to know your
company's primary NAICS code. Could you please provide that?

TURN 4, STEP 8, user (underwriter) chat:
I do not know.

TURN 4, STEP 9, assistant (internal assistant) action:
TOOL CALLS: [{'name': 'get_table_descriptions', 'args': {}, 'id':
'call_Zn8kwoWwSdDbMES5zATzjwHA', 'type': 'tool_call'}]

TURN 4, STEP 10, assistant (tool) chat:
{"naics": "table with 2022 6-digit NAICS codes", "naics_2digit": "table
with 2022 2-digit NAICS codes", "naics_2022_2017": "table that maps 2022
NAICS codes to 2017 NAICS codes", "naics_2012_2017": "table that maps
2012 NAICS codes to 2017 NAICS codes", "small_business_definition":
"table that defines what a 'small' business is by its 6-digit 2012 NAICS
code and annual revenue in US dollars", "lobs": "table of lines of
business being offered for small businesses", "states": "table of US
state names and abbreviations", "appetite_guide": "table that defines
appetite for writing a policy for a small business depending on features
of the business such as business classification"}

TURN 4, STEP 11, assistant (user-facing assistant) chat:
Could you describe the company's primary business activities (e.g. the
products or services it provides)? That will let me map it to the correct
```

```
NAICS code and determine the standard cyber limit.

TURN 5, STEP 12, user (underwriter) chat:
Outpatient care: sleep, pain, diet counseling.

TURN 5, STEP 13, assistant (internal assistant) action:
TOOL CALLS: [{'name': 'read_query', 'args': {'query': "SELECT code,
title FROM naics WHERE code LIKE '621%'"}, 'id':
'call_ELye3JMtvfjulBIAQHr8K5yd', 'type': 'tool_call'}]

TURN 5, STEP 14, assistant (tool) chat:
[{"Code": 621111, "Title": "Offices of Physicians (except Mental Health
Specialists)"}, {"Code": 621112, "Title": "Offices of Physicians, Mental
Health Specialists"}, {"Code": 621210, "Title": "Offices of Dentists"},
{"Code": 621310, "Title": "Offices of Chiropractors"}, {"Code": 621320,
"Title": "Offices of Optometrists"}, {"Code": 621330, "Title": "Offices
of Mental Health Practitioners (except
...Truncated for length...
8, "Title": "All Other Outpatient Care Centers"}, {"Code": 621511,
"Title": "Medical Laboratories"}, {"Code": 621512, "Title": "Diagnostic
Imaging Centers"}, {"Code": 621610, "Title": "Home Health Care Services"},
{"Code": 621910, "Title": "Ambulance Services"}, {"Code": 621991,
"Title": "Blood and Organ Banks"}, {"Code": 621999, "Title": "All Other
Miscellaneous Ambulatory Health Care Services"}]

TURN 5, STEP 15, assistant (internal assistant) action:
TOOL CALLS: [{'name': 'read_query', 'args': {'query': "SELECT naics_code,
max_revenue FROM small_business_definition WHERE naics_code LIKE '621%';
"}, 'id': 'call_tRrggv8ACgckUpGDNFo4KI6V', 'type': 'tool_call'}]

TURN 5, STEP 16, assistant (tool) chat:
Error: ToolException('Error executing tool read_query: SQLite error:
no such column: naics_code')
 Please fix your mistakes.

TURN 5, STEP 17, assistant (internal assistant) action:
TOOL CALLS: [{'name': 'get_table_schema', 'args': {'table_name':
'small_business_definition'}, 'id': 'call_lkRTGgWQseW7lSSXIEUhJK6K',
'type': 'tool_call'}]

TURN 5, STEP 18, assistant (tool) chat:
[{"cid": 0, "name": "index", "type": "INTEGER", "notnull": 0,
"dflt_value": null, "pk": 0}, {"cid": 1, "name": "NAICS Codes", "type":
"INTEGER", "notnull": 0, "dflt_value": null, "pk": 0}, {"cid": 2, "name":
"Size Standards in millions of dollars", "type": "REAL", "notnull": 0,
"dflt_value": null, "pk": 0}, {"cid": 3, "name": "Size standards in
number of employees", "type": "REAL", "notnull": 0, "dflt_value": null,
"pk": 0}]

TURN 5, STEP 19, assistant (internal assistant) action:
TOOL CALLS: [{'name': 'read_query', 'args': {'query': 'SELECT "NAICS
Codes","Size Standards in millions of dollars" FROM
small_business_definition WHERE "NAICS Codes"=621498;'}, 'id':
'call_igqqWCW27Z249BtsE1MQZLOI', 'type': 'tool_call'}]

TURN 5, STEP 20, assistant (tool) chat:
[{"NAICS Codes": 621498, "Size Standards in millions of dollars": 20.5}]

TURN 5, STEP 21, assistant (user-facing assistant) chat:
FINAL ANSWER: For the Cyber LOB on an outpatient care center (NAICS
621498, which falls under the 3-digit 621 \Ambulatory Health Care
Services" category), All National Insurance's standard recommendation
is $3 million per-occurrence and $5 million aggregate.

Rationale:
```

```
156   • Underwriting guidelines designate small businesses in NAICS 621 as
157   eligible for Cyber coverage.
158   • Cyber limits for those classes are set higher|$3 M per-occurrence and
159   $5 M aggregate|due to the sensitive customer data handled by health-care
160   service providers.
161   • Your company's $6.7 M revenue is below the $20.5 M small-business
162   threshold for NAICS 621498, so it qualifies as a small business.
163
164   MOST PROBABLE BRANCH:
165   is end<|eot_id|>
```

**WebShop, 19 branches, max logprob -6.72, completed=True, goal='Product selection and comparison intent'**

```
1    TURN 1, STEP 1, user chat:
2    You are web shopping.
3    I will give you instructions about what to do.
4    You have to follow the instructions.
5    I will give you an observation and a list of available actions, you have
6    to respond an action based on the state and instruction.
7    You can use the following actions: [{'name': 'search', 'description':
8    search on the web with keywords'}, {'name': 'click', 'description':
9    'click the value in webpa
10   ...Truncated for length...
11    the buttons in clickables.
12   If the action is not valid, perform nothing.
13   Keywords in search are up to you, but the value in click must be a
14   value in the list of available actions.
15   Remember that your keywords in search should be carefully designed.
16   WebShop Instruction: i'm hoping to find non-toxic false teeth that
17   are made out of high quality soft silicone, and price lower than
18   40.00 dollars Search
19
20   TURN 1, STEP 2, assistant chat:
21   Thought:
22   I think I should use the search bar to look for non-toxic false teeth
23   made of high-quality soft silicone within my price range.
24
25   Action:
26   search[non-toxic false teeth high-quality soft silicone price < 40.00]
27
28   TURN 2, STEP 3, user chat:
29   Instruction: i'm hoping to find non-toxic false teeth that are made out
30   of high quality soft silicone, and price lower than 40.00 dollars Back
31   to Search Page 1 (Total results: 50) Next > B09KRS2GPJ Velaurs Fangs,
32   Soft Silicone False Teeth Vivid Retractable for Parties for Halloween
33   arty for Mandibular Teeth for Lay Pops $6.03 B09KRTXY77 bizofft False
34   Teeth, Soft Silicone Reusable Fangs with Stora
35   ...Truncated for length...
36   ture Bath Case False Teeth Storage Box Container with Basket Net
37   Container Holder for Travel, Retainer Cleaning(Dark Blue) $8.19
38   B093BDB92R Mouth Guard, Silicone Teeth Guard, Sleep Anti-molar Aid
39   Braces, for Bruxism Clenching, Teeth Grinding, Stoping Bruxism, 1 Pack
40   $9.69 B07Q2QFN6S 8 PCS Silicone Mouth Guard for Teeth Clenching
41   Grinding Moldable Dental Night Guards Bite Sleep Aid By Giveme5 $9.99
42
43   TURN 2, STEP 4, assistant chat:
44   Thought:
45   I should click on 'B09KRS2GPJ' as it seems to be a non-toxic false teeth
46   option made of high-quality soft silicone.
47
48   Action:
49   click[B09KRS2GPJ]
```

```
TURN 3, STEP 5, user chat:
Instruction: i'm hoping to find non-toxic false teeth that are made out
of high quality soft silicone, and price lower than 40.00 dollars Back
to Search < Prev Velaurs Fangs, Soft Silicone False Teeth Vivid
Retractable for Parties for Halloween Party for Mandibular Teeth for Lay
Pops Price: $6.03 Rating: N.A. Description Features Reviews Buy Now

TURN 3, STEP 6, assistant chat:
Thought:
I should click on 'Buy Now' to purchase the non-toxic false teeth made
of high-quality soft silicone.

Action:
click[Buy Now]

MOST PROBABLE BRANCH:
 based the chat logsI
```

**KB+OS+SQL, 26 branches, breaklines, max logprob -0.67, completed = True, goal = 'Filesystem analysis and content summarization'**

```
TURN 1, STEP 1, user chat:
Your goal is to implement the operations in OS system and answer to the
question proposed by me. You have access to the following actions:
[{'name': 'bash', 'description': 'standard bash tool to run command in
OS system'}, {'name': 'answer', 'description': 'if you have the answer,
call this function'}]. Please generate the exact input for actions at
each step. If the output is too long, I will truncate it. The truncated
output is not complete. You have to deal with the truncating problem by
yourself. Attention, your bash code should not contain any input
operation, follow the format with ```bash
#put your bash code here
```
You need to find out how many unique extensions of files are there in
the current directory.

TURN 1, STEP 2, assistant chat:
Think: First, I need to list all of the files in the current directory.

Act: bash

```bash
ls -p | grep -v /
```

TURN 2, STEP 3, user chat:
bin
docs.pdf
docs2.pdf
lib
lib32
lib64
libx32
more.data.txt
music1.mp3
music2.wav
sample1.jpg
sample3.txt
sample4.mp4
sample5.avi
sbin
]0;root@9ed33d69bda1: /root@9ed33d69bda1:/#
```

```
TURN 2, STEP 4, assistant chat:
Think: Now I have all the files. I need to extract the extensions.

Act: bash

```bash
ls -p | grep -v / | awk -F . '{if (NF>1) {print $NF}}'
```

TURN 3, STEP 5, user chat:
pdf
pdf
txt
mp3
wav
jpg
txt
mp4
avi
]0;root@9ed33d69bda1: /root@9ed33d69bda1:/#

TURN 3, STEP 6, assistant chat:
Think: Now, we have all the file extensions. Now we need to sort and
find unique extensions.

Act: bash

```bash
ls -p | grep -v / | awk -F . '{if (NF>1) {print $NF}}' | sort -u
```

TURN 4, STEP 7, user chat:
avi
jpg
mp3
mp4
pdf
txt
wav
]0;root@9ed33d69bda1: /root@9ed33d69bda1:/#

TURN 4, STEP 8, assistant chat:
Think: Now, we have a list of unique file extensions in the current
directory. The final step is to count these extesions.

Act: bash

```bash
ls -p | grep -v / | awk -F . '{if (NF>1) {print $NF}}' | sort -u | wc -l
```

TURN 5, STEP 9, user chat:
7
]0;root@9ed33d69bda1: /root@9ed33d69bda1:/\#

TURN 5, STEP 10, assistant chat:
Think: Now we get the answer. There are 7 unique extensions in the
current directory.

Act: answer(7)

MOST PROBABLE BRANCH:
  <|end_of_text|>
```

