# OpenReview forum: "Unsupervised Evaluation of Multi-Turn Objective-Driven Interactions"
_ICLR.cc/2026/Conference — Submitted to ICLR 2026_

### Official Review · Reviewer_hA7u · 2025-10-30

**Soundness:** 2
**Presentation:** 2
**Contribution:** 2
**Rating:** 2
**Confidence:** 4

**Summary:**

This paper proposes an unsupervised framework for evaluating large language models (LLMs) in multi-turn, goal-oriented interactions without relying on human labels or LLM judges. The framework has three modules: a) Goal Identification uses LLM summarization and embedding-based clustering to discover latent user intents in unlabeled dialogues; b) Completion Detection fine-tunes a Llama 3 8B model to predict whether a conversation has achieved its goal using a special <|end|> token; c) Response Trees – introduces a novel visualization of model uncertainty based on the branching structure of possible next responses. Experiments across datasets such as LMSYS-Chat-1M, Code-Feedback, Insurance, WebShop, and SQL+OS+KB show that the small completion model can approach or even outperform large LLM judges on binary goal-completion classification (F1 up to 0.99). However, the Response Tree component is conceptual and not empirically validated.

**Strengths:**

The paper focuses on a highly relevant and underexplored challenge in the current LLM research landscap: how to evaluate multi-turn, goal-oriented agents without relying on costly human annotations or unstable LLM-as-a-judge paradigms. As LLMs increasingly act as autonomous or semi-autonomous agents engaging in extended task-driven interactions, scalable and objective evaluation becomes a bottleneck for both research and deployment.

**Weaknesses:**

1. The “unsupervised” claim is misleading: all three modules depend on LLM summarization, synthetic labeling, or fine-tuning.
2. The negative sampling strategy (random truncation) does not accurately simulate real incomplete dialogues.
3. The goal-identification module requires embedding and clustering the entire dataset at once, relying on iterative global merging and cross-cluster sampling. This design makes the method inherently batch-based, it cannot process streaming data without full re-clustering (e.g., in online systems).
4. The Response Tree idea is not evaluated quantitatively; no correlation analysis links branching structure to uncertainty or correctness.
5. The overall empirical contribution is confined to the completion detector, leaving the framework’s other parts remain speculative.

**Questions:**

Please refer to Weaknesses.

---

> ### Author Response · Authors · 2025-11-22
> **Response to Reviewer hA7u**
>
> We thank the reviewer for their response, which highlights the relevance of our work to current LLM research and the benefits our evaluation approach provides, especially as LLMs act in challenging-to-evaluate agentic contexts to achieve specific user goals. We address the reviewer’s concerns in detail below.
>
> **"Unsupervised" wording**
> Several reviewers raised concerns about the “unsupervised” phrasing in the title. We propose changing the paper’s title to use the phrase “weakly supervised” for clarity. We have also changed “unsupervised” to “weakly supervised” throughout the paper.
>
> **Use of truncated samples to simulate incomplete interactions**
> We applied the negative sampling strategy because there are no publicly available datasets of both complete and incomplete dialogues. Although one would expect to encounter different types of incomplete dialogues in practice, truncated conversations from high-quality fine-tuning datasets, which primarily  are a reasonable simulation in the absence of purpose-built datasets for completion labeling.
>
> **Clustering is inherently batch-based**
> As presented, this is true. Future work could extend the goal-identification module, studying its performance when, for example, a partial dataset is clustered and the remaining samples are assigned to the preexisting clusters. This is a good practical extension of our work, and we thank the reviewer for their suggestion confirming it is a topic of interest.
>
> **Response tree evaluation and empirical contribution**
> To respond to reviewer feedback, we have expanded on the mathematical justification for the use of response trees to characterize LLM uncertainty. We emphasize that the response tree is an empirical measurement of uncertainty: it efficiently captures different potential responses to a given prompt.
> Thanks to reviewers’ feedback, we expanded on our justification in Section 4.3 to add a probabilistic argument for how the response tree reflects LLM uncertainty in the correct response to a given prompt. We reproduce this argument below:
>
> Suppose an LLM response tree for prompt $p$ has one branch representing response $r$, which will be generated with probability $\geq 1-\alpha$. This response will be consistently generated in response to prompt $p$ \footnote{It is possible for $r$ to be an incorrect response to $p$; this would not be detectable by our metric, and would occur if the LLM is highly confident in an incorrect answer.}.
> Now suppose an LLM response tree has $n$ branches representing responses $r_1,\dots,r_n$, such that $r_1,\dots,r_n$ make up at least $1-\alpha$ of the probability mass for the distribution of responses to prompt $p$.  If one were to sample many responses, one would expect to see $r_1,\dots,r_n$ in the resulting samples. Since $r_i \neq r_j$, the increased number of branches reflects uncertainty in the response to prompt $p$. Assuming there is one correct response, a large number of leaf nodes corresponds to an increased likelihood that an incorrect response will be generated.
> We have also updated Section 4 to include the Adjusted Rand Index for measuring cluster stability, which shows empirically that our clustering algorithm outperforms an LLM judge baseline.
>
> We thank the reviewer for their suggestions, which led us to expand on the empirical evaluation of our approach.

---

### Official Review · Reviewer_wVNW · 2025-11-01

**Soundness:** 3
**Presentation:** 3
**Contribution:** 2
**Rating:** 6
**Confidence:** 3

**Summary:**

This paper introduces three unsupervised metrics for evaluating multi-turn objective-driven interactions between users and LLMs: (1) goal identification through LLM-guided clustering that combines k-means with LLM-generated labels, (2) goal completion detection using fine-tuned LLMs that predict an "end tag" for completed conversations, and (3) response uncertainty quantification via response trees that enumerate probable LLM outputs. The authors validate their approach on diverse datasets including LMSYS-Chat-1M, code-feedback, insurance underwriting, WebShop, and technical assistance tasks (SQL+OS+KB), demonstrating that 8B fine-tuned models can match or exceed 70B LLM judges on completion labeling.

**Strengths:**

1. The core ideas—unsupervised goal clustering via LLM+k-means hybridization, completion detection via distributional fine-tuning, and uncertainty quantification via response trees—are highly original.

2.  The methods address a real and pressing need in enterprise AI development. The ability to perform evaluation without labels or human judges is a substantial practical advance.

**Weaknesses:**

1. The method assumes "a majority of interactions are complete" to train the completion detector, which fundamentally undermines its unsupervised nature and creates a validation problem.

2. The pipeline likely depends on embedding model, initial k, prompt phrasing for merges/labels, and decoding hyperparameters. Robustness to these choices is not fully demonstrated.

**Questions:**

1. How do you address the fundamental circularity in assuming most interactions are complete to detect completion? Could you validate on datasets with known failure rates?

2. Why does completion detection fail so dramatically on Code-Feedback (F1=0.21)? This seems to invalidate the approach for follow-up conversations.

3. What is the computational cost vs. using LLM judges? Is fine-tuning + inference cheaper?

4. In Algorithm 1, an LLM is prompted to decide if two clusters should be merged. Given the paper's valid critiques of LLM judge instability, what steps were taken to ensure this "merge" decision is stable and reliable? Did you observe any of the instability (e.g., order sensitivity) that you found in the LLM-only baseline?

---

> ### Author Response · Authors · 2025-11-22
> **Response to Reviewer wVNW**
>
> We thank the reviewer for their careful review, both acknowledging the novelty and relevance of our work and raising several interesting questions, which we discuss below.
>
> **Completeness**
> We see the assumption that most interactions are complete as being similar to a common assumption in outlier detection; that almost all data points are not outliers. Although the method we propose is not a traditional outlier detection method, it has one major similarity: the use of a dataset that may contain a few incomplete samples, in the same sense that a dataset used to train an outlier detection model may contain a few outliers.
> One limitation we are aware of is that this paper does not provide a theoretical analysis of this approach. While such an analysis is extremely desirable, it remains technically challenging as we are not aware of any theoretical results on the performance of fine-tuning algorithms. In Section 5.3 we added a paragraph expanding on this challenge:
>
> **Clustering robustness to hyperparameter choices**
> We acknowledge these limitations. An in-depth study of how prompt formatting, hyperparameters and other design choices contribute to the performance of our clustering algorithm would be useful, however it is out of the scope of this work. Thanks to several reviewers’ comments, we did expand on our analysis of clustering stability and now report Adjusted Rand Index numbers for all datasets, which we use to quantify how well two runs of the same clustering algorithm agree with each other. We show that our method is significantly more stable than an LLM-as-a-judge approach.
>
> One natural follow-up question is whether we can relate the performance of this approach to the percentage of incomplete conversations in the training dataset. However, to conduct this analysis one would need to relate the prevalence of certain tokens in a dataset to the probability of generating said tokens after fine-tuning on the dataset. Unfortunately, theoretical analysis of fine-tuning performance remains an understudied question; thus analysis of this approach is out of the scope of this paper, but is an excellent direction for future work.
>
> Finally, although a theoretical analysis of LLM fine-tuning in this context is beyond the scope of this paper, we consider it an excellent direction for future work.
>
> **Why does our completion approach perform poorly on Code-Feedback?**
> Because the structure of the Code-Feedback dataset is
>  Prompt 1
> Response 1 (fully answers prompt 1)
> Prompt 2 (Follow-up)
> Response 2 (fully answers prompt 2)
> …and so on
> There isn’t a natural endpoint, as another follow-up is always plausible. This is a limitation of the statistical modeling approach we use to label completion. On other datasets reflecting agentic LLM use cases, which are often challenging to evaluate due to the distribution shift in agentic interaction logs, our method performs much more reliably. We included the Code-Feedback results to demonstrate this phenomenon.
>
> **Computational cost**
> It seems reasonable that, depending on the number of different datasets (with different distributions) being analyzed, fine-tuning + inference on a small (8 billion parameter) model for each distribution would yield significant computational cost savings compared to evaluating each (potentially long) sample with a modern LLM judge. Because there are multiple factors that contribute to the cost of fine-tuning and interference, such as dataset size, context length and hardware optimization, as well as factors contributing to the cost of LLM judges (number of calls, prompt length, sample length), we did not provide a numeric comparison.
>
> **Clustering and LLM merge**
> We designed our cluster merging prompt to minimize the impact of uncertainty in LLM judges. Specifically, we present several randomly selected samples from each cluster and several samples from different clusters, which serve as a negative example. This turns a poorly-grounded question (“are the topics in these two clusters similar?”) into a more well-grounded question (“are the topics in these two clusters similar, in comparison to the topics in other, dissimilar clusters?”). We acknowledge that there are still opportunities for phenomena such as a bias towards the first choice, or a bias induced by the wording of our prompt, however we did not observe these instabilities in practice. We provide full details for our cluster merging approach, including an example, in  Appendix B.
> Additionally, another reviewer suggested reporting the Adjusted Rand Index, a measure of cluster stability. We added these results in Section 5.2, showing our algorithm significantly outperforms GPT-4.1 as an LLM judge.

---

### Official Review · Reviewer_Xmjj · 2025-11-01

**Soundness:** 2
**Presentation:** 2
**Contribution:** 2
**Rating:** 2
**Confidence:** 4

**Summary:**

This paper proposes the first unsupervised metrics to evaluate objective-driven AI–human interactions, addressing the limits of human annotation, bespoke checks, and unreliable LLM judges on complex unlabeled data.
Leveraging statistical signals and fine-tuned LLMs to handle distribution shifts, the metrics label user goals, assess goal completion, and estimate LLM uncertainty, with validation on open-domain and task-specific datasets.

**Strengths:**

- The paper addresses key shortcomings in current LLM evaluation methods.
- It presents extensive experiments demonstrating the effectiveness of the proposed approaches.

**Weaknesses:**

- The overall paper structure could be improved. A concise introduction is fine, but it should still highlight the main points to orient the reader.
- The selected attributes require a clear rationale for why they are important for evaluating LLM responses.
- The metrics used to measure them also need clearer explanation and justification: how well do these metrics reflect the degree or quality of the attributes?
- The title is potentially misleading. What exactly is meant by “unsupervised”? Is a fine-tuned model used for completion evaluation?
- How do you clearly justify the claim that a larger number of leaves implies greater uncertainty in LLM responses?

**Questions:**

Please see the Weaknesses section.

---

> ### Author Response · Authors · 2025-11-22
> **Response to Reviewer Xmjj**
>
> We thank the reviewer for their careful feedback, which both acknowledges that this work is highly relevant to current LLM evaluation and validated on diverse datasets, and identifies several weaknesses in the paper’s presentation that we have corrected in the revised version.
>
> **Overall Paper Structure Improvements**
> We thank the reviewer for their feedback on the paper structure. We have expanded the abstract and introduction to more clearly introduce our work. Specifically, we changed:
> We introduce three LLM judge-free metrics for labeling user goals, goal completion, and LLM uncertainty. Our metrics leverage implicit properties of objective-driven interaction data and can be applied under reasonable assumptions without labels or ground-truth responses.
> To
>
> We introduce three LLM judge-free metrics for labeling user goals, goal completion, and LLM uncertainty. Our metrics are designed for frontier LLMs where errors are rare and conversations are typically of high quality. We leverage these assumptions and implicit properties of objective-driven interaction data to introduce the first set of weakly supervised metrics for objective-driven interactions. Specifically, we present: a stable clustering algorithm for identifying user goals; the use of small fine-tuned LLMs to adapt to distributional shifts; and a novel \textit{response tree} structure representing multiple possible LLM responses, which can be efficiently constructed and used to understand the conditional distribution of LLM responses given a specific prompt. We apply these tactics to develop metrics for labeling user goals, measuring goal completion, and quantifying LLM uncertainty: evaluation axes of interest to agentic system developers.
>
> **Clarifying Rationale for Metrics**
> In the introduction and Section 3 we discuss why the selected attributes are of interest in an industry context; companies providing LLM services for their customers or employees (“users”) are interested in what topics their LLMs are assisting users with, whether users are successfully completing tasks with LLM assistance, and LLM uncertainty, which suggests users could encounter variation in LLM responses over multiple trials. Because these attributes are difficult to measure directly, we provide logical and mathematical arguments in Section 4 for how the concrete attributes we measure relate to these high-level descriptions.
>
> We also added additional metrics for empirical justification: reporting the Adjusted Rand Index, a measure of clustering agreement, to show empirically how our clustering algorithm outperforms LLM-as-a-judge; and computing p-values for the results of our completion labeling approach. Additionally, we  added a mathematical justification relating the response tree size to LLM uncertainty in Section 4.3.
>
> **Unsupervised Phrasing**
> Several reviewers raised concerns about the “unsupervised” phrasing in the title. We propose changing the paper’s title to use the phrase “weakly supervised” for clarity. We have also changed “unsupervised” to “weakly supervised” throughout the paper.
> We do use a fine-tuned small language model, which we treat as a model of the distribution of interactions, for completion evaluation. Fine-tuning trains the model to generate the end tag when prompted with a complete conversation; and to generate a plausible conclusion to the conversation, followed by the end tag, when prompted with an incomplete conversation. The use of a fine-tuned model to represent a specific dataset’s distribution is one of the novel contributions of our work.
>
> **Response Trees and their Relationship to Uncertainty**
> We can justify this claim by a probabilistic argument, which we have added in Section 4.3:
> We justify this metric via a probabilistic argument. Suppose an LLM response tree for prompt $p$ has one branch representing response $r$, which will be generated with probability $\geq 1-\alpha$. This response will be consistently generated in response to prompt $p$ \footnote{It is possible for $r$ to be an incorrect response to $p$; this would not be detectable by our metric, and would occur if the LLM is highly confident in an incorrect answer.}.
>
> Now suppose an LLM response tree has $n$ branches representing responses $r_1,\dots,r_n$, such that $r_1,\dots,r_n$ make up at least $1-\alpha$ of the probability mass for the distribution of responses to prompt $p$.  If one were to sample many responses, one would expect to see $r_1,\dots,r_n$ in the resulting samples. Since $r_i \neq r_j$, the increased number of branches reflects uncertainty in the response to prompt $p$. Assuming there is one correct response, a large number of leaf nodes corresponds to an increased likelihood that an incorrect response will be generated.

---

> > ### Comment · Reviewer_Xmjj · 2025-11-23
> >
> > Thank you to the authors for their time and effort in responding. However, I did not find additional evidence and am not fully convinced by the rationale behind the evaluation and the proposed concept. Therefore, I will maintain my original score.

---

> > > ### Author Response · Authors · 2025-11-27
> > > **Request for clarification on reviewer concerns**
> > >
> > > **Concerns about additional evidence**
> > >
> > > We apologize if these updates weren't clear in our first response. In our revised paper, we added additional evaluation metrics in **Section 5**.
> > > 1. To evaluate our completion labeling, we computed p-values (Table 3) to determine whether we can reject the null hypothesis “our completion labeling does not have higher accuracy than an LLM judge”, showing a statistically significant difference on all datasets except SQL+OS+KB, where both methods are highly accurate.
> > > 2. To support our claim that our clustering algorithm is more stable than an LLM judge, we report the Adjusted Rand Index, a measure of the agreement between two clustering assignments, in Table 2. We find our algorithm significantly outperforms GPT-4.1 as an LLM judge.
> > >
> > > **Concerns about rationale**
> > >
> > > Can you please elaborate on the specific weaknesses you find in our evaluation? Other reviewers have noted our “core ideas … are highly original” and are a "substantial practical advance” (Reviewer wVNW) and that our "ablation studies effectively justify key design choices” (Reviewer UJoU). We are, of course, interested in strengthening our paper and would welcome this feedback.

---

### Official Review · Reviewer_UJoU · 2025-11-07

**Soundness:** 3
**Presentation:** 3
**Contribution:** 2
**Rating:** 6
**Confidence:** 3

**Summary:**

This paper addresses a critical and timely problem in the deployment of LLMs in enterprise settings: the evaluation of multi-turn, objective-driven interactions (e.g., customer service, task-oriented agents) without relying on human annotations or unreliable LLM-as-a-judge paradigms. The authors propose a novel suite of three unsupervised metrics:
LLM-Guided Clustering: An algorithm that combines k-means on text embeddings with LLM-based labeling and merging to automatically discover and label user goals from unlabeled conversation data.
Interaction Completeness: A method that fine-tunes a small LLM on a dataset of (mostly) completed conversations, tagged with a special "end" token, to predict whether a given interaction is complete. Incomplete interactions are flagged as outliers.
Response Uncertainty: A framework using "response trees" to quantify the LLM's uncertainty during a conversation by exploring the branching factor and log-probabilities of potential responses, serving as a proxy for interaction quality and potential error likelihood.
The methodology is empirically validated on a range of datasets, including open-domain chats and specialized, tool-using interactions. The key result is that their approach, often using a fine-tuned 8B parameter model, can match or exceed the performance of a much larger (70B) LLM judge on the task of completion labeling, demonstrating the potential for efficient, specialized evaluation.

**Strengths:**

Novelty and Practical Impact: The core idea of moving beyond LLM judges and human references to unsupervised, statistically-grounded metrics is highly novel and addresses a significant pain point in real-world AI system development. The potential for online monitoring and resource-saving interventions is a compelling practical contribution.
Holistic Evaluation Framework: The paper doesn't propose a single metric but a suite of three complementary metrics that address different aspects of an interaction (goal, completion, uncertainty), providing a more comprehensive evaluation toolkit.
Strong Empirical Validation: The paper is thorough in its experimentation, using multiple diverse datasets (LMSYS, Code-Feedback, Insurance, WebShop, etc.) to demonstrate the generality of the approach. The ablation studies (e.g., "No end tag," comparison of fine-tuned vs. base models) effectively justify key design choices.
Efficiency Focus: The successful use of small (8B parameter) models, fine-tuned for specific distributions, is a major strength. It challenges the prevailing trend of using ever-larger models as judges and opens avenues for cost-effective, scalable evaluation.
Clarity of Core Methodology: The underlying concepts for the completeness and uncertainty metrics—modeling the token distribution of completed interactions and constructing response trees—are well-explained and theoretically sound.

**Weaknesses:**

Reliance on Key Assumptions: The methodology rests on two strong assumptions that may not always hold in practice: a single user goal per interaction and that failures are "rare." The performance degradation on Code-Feedback, where follow-up questions violate the "single well-defined end" assumption, highlights this fragility. The applicability in noisier, real-world environments with frequent failures is not fully established.
Limited Statistical Rigor: While the results are promising, the statistical significance of the improvements is not tested. Table 2 reports accuracy and F1 scores, but without confidence intervals or statistical tests, it's difficult to gauge the robustness of the claims, especially on smaller datasets like Insurance (n=380).
Writing and Presentation Issues:
Inconsistent Metric Naming: The three core metrics are introduced as labeling user goals, measuring goal completion, and quantifying LLM uncertainty. However, in the methodology and experiments, they are referred to as "clustering," "completion labeling," and "response trees." A consistent, clear naming scheme would improve readability.
Undefined "Stability": The clustering algorithm is praised for its "stability," but this term is not quantitatively defined. Figure 4 is visual and compelling, but a quantitative measure of cluster stability (e.g., Adjusted Rand Index between runs) would strengthen the claim.
Supervision in "Unsupervised" Learning: The approach requires a dataset where "the majority of interactions are complete" to fine-tune the completeness model. While not requiring manual labels, this does require a curated set of complete conversations, which is a form of weak supervision. The term "unsupervised" should be nuanced to acknowledge this data curation requirement.

**Questions:**

Assumptions and Generalizability: How would your completeness metric perform in a production environment where user interactions frequently contain multiple, interleaved goals or where the failure rate is significantly higher than in your training data? Are there strategies to make the approach more robust to such violations of its core assumptions?
Statistical Significance: Given the varying sizes of your datasets, have you performed any statistical significance testing (e.g., bootstrapping) on the results in Table 2? This would help clarify whether the observed performance differences, particularly on smaller datasets like Insurance, are reliable.
Clustering Stability Metric: You state your clustering algorithm produces "highly stable clusters." Beyond the confusion matrices, did you calculate a quantitative stability metric like the Adjusted Rand Index between different runs? If so, what were the results?
Response Tree Practicality: The response tree is a powerful conceptual tool for quantifying uncertainty. However, the computational cost of building these trees for long, complex conversations must be non-trivial. Could you comment on the computational overhead of this method and its feasibility for near-real-time monitoring?
Defining "Completion": The "end" tag is a clever heuristic, but it conflates syntactic and semantic completion. A conversation could end with the user giving up. Does your method have a way to distinguish between a successfully resolved goal and an abandoned one, or is it solely based on the model's learned distribution of "end-of-dialogue" tokens?

---

> ### Author Response · Authors · 2025-11-22
> **Response to Reviewer UJoU**
>
> We thank the reviewer for their careful and in-depth response highlighting both the positives and weaknesses of this work. We address their concerns in-depth below.
>
> **Reliance on Key Assumptions**
> The methodology rests on two assumptions that may not always hold in practice: a single user goal per interaction and that failures are "rare”. We argue that, although many user-LLM interactions do not follow these assumptions, they are reasonable in the domain of agentic assistants or business-oriented LLM conversations. Focusing on the limited scope of single-task goal-oriented conversations with highly capable LLMs has allowed us to develop novel algorithms that take advantage of the attributes of these datasets to efficiently evaluate LLM-human interactions without human annotation.
> We focus on the case of rare, potentially-unknown failures because if failures are common or one has prior knowledge of the types of failures that might occur, they can be detected by inspection or designing custom metrics. It is true that our method is strongest when failures are rare; however this is also the most difficult setting for detecting them by other methods.
>
> In Section 5.3 we added a paragraph explaining the challenges of a theoretical analysis of our completion labeling method.
>
> **Limited Statistical Rigor**
> Table 3 (in the revised version) reports aggregated accuracy, precision, recall, and F1 scores for completion labeling. One possible statistical test is whether we can reject the null hypothesis “our completion labeling does not have higher accuracy than an LLM judge”. We have computed p-values for this test in Section 5.3, showing a statistically significant difference between the LLM judge baseline and our approach on all datasets except SQL+OS+KB, where both methods are highly accurate.
>
> **Writing and Presentation Issues**
> We thank the reviewer for their close attention to detail. In Section 3, we clarified our definition of LLM uncertainty in Section 3 and changed “Goal Identification” to “Goal Clustering” and “Response uncertainty” to “Response tree” in several places.
>
> **Undefined "Stability"**
> We thank the reviewer for this suggestion; we agree this is an excellent metric to report. We updated Section 5.2 with new results showing our algorithm significantly outperforms GPT-4.1 as an LLM judge.
>
> **Supervision in "Unsupervised" Learning**
> Several reviewers raised concerns about the “unsupervised” phrasing in the title. We propose changing the paper’s title to use the phrase “weakly supervised” for clarity. We changed “unsupervised” to “weakly supervised” throughout.
>
> **Assumptions and Generalizability**
> The main reason we made the assumption that the user has a single goal is to simplify the design of our clustering algorithm. We note that there is a difference between a user having multiple sequenced goals which occur together, such as writing, proofreading and sending a document via email (which could be treated as one goal of “document preparation”); and a user having multiple unrelated goals (for example, first asking a coding agent to write test cases for one piece of code, and then asking for assistance deploying an unrelated webpage). If users have sequenced goals that occur together, we can treat them as a single larger goal. If users have multiple unrelated goals in one conversation, we expect this to degrade the performance of our clustering algorithm as there is no longer a single clear category for each conversation.
> For this paper, which introduces novel metrics for conversation evaluation without human annotation, generalizing our method to settings where our assumptions are violated is out of scope. However, we thank the reviewer for their interest in this topic, as we agree it is an excellent direction for future work.
>
> **Response Tree Practicality**
> To avoid this situation, we truncate the response tree generation to  control the computational complexity by only branching on tokens where $n>1$ tokens have probability $\geq\alpha$, where $\alpha$ is some threshold value generating many low-probability responses. In Section 4.3 we added the sentence “In practice, we constrain the number of branches using $\alpha$ and limit their depth to reduce the computational complexity of generating the tree.” to clarify this process.
>
> **Defining "Completion"**
> The method we propose is based on the model's learned distribution of "end-of-dialogue" tokens, so if many conversations are incomplete or end with the user giving up, our method will not be able to detect this.
> We explore this limitation with the Code-Feedback dataset, which contains samples formatted as “question 1”, “response 1”, “follow-up”, “response to follow-up”, “follow-up”, “response to follow-up”, etc. and thus does not have a distinct ending (another follow-up is always possible). In comparison, the other datasets, most of which represent agentic interactions, do have natural endpoints where a task is complete.

---

### Author Response · Authors · 2025-11-22
**Global Response to All Reviewers**

We thank the reviewers for their thoughtful comments, many of which emphasized the novelty and applicability of our work. Our reviewers made multiple insightful suggestions which have improved the clarity and statistical rigor of our paper. We made several important changes, which we summarize here.

**Unsupervised vs weakly supervised**
Several reviewers raised concerns about the “unsupervised” phrasing in the title. We propose changing the paper’s title to use the phrase “weakly supervised” for clarity. We have also changed “unsupervised” to “weakly supervised” throughout the paper.

**Statistical testing of completion results**
We report aggregated accuracy, precision, recall, and F1 scores for completion labeling, treating truncated interactions as ground-truth “incomplete” samples and full interactions as ground-truth “complete” samples. Thanks to reviewer feedback, we computed p-values to determine whether we can reject the null hypothesis “our completion labeling does not have higher accuracy than an LLM judge”. These results show a statistically significant difference on all datasets except SQL+OS+KB, where both methods are highly accurate

**Additional metrics of clustering stability**
One reviewer suggested we report the Adjusted Rand Index, a measure of the agreement between two clustering algorithms (or two runs of the same nondeterministic algorithm). This index is 1 when the two algorithms agree perfectly, and 0 when their agreement is no better than random chance. We computed these values for each dataset, finding that our algorithm significantly outperforms GPT-4.1 as an LLM judge. We updated Section 5.2 with an additional paragraph and table presenting these results.

**Justification of the response tree and uncertainty**
We expanded on our justification in Section 4.3 to add a probabilistic argument for how the response tree reflects LLM uncertainty in the correct response to a given prompt.

---

### Author Response · Authors · 2025-12-03
**Summary Message to the Area Chair**

## Our Paper
We develop novel, weakly supervised evaluation metrics for objective-driven interactions, including those in difficult agentic settings where LLMs may interact with custom tooling or business logic, resulting in out-of-distribution interactions.
We propose three metrics to evaluate these interactions without human annotation or LLM judges.

**LLM-Guided Clustering**:
A novel algorithm that combines LLM summarization with k-means on embedded summaries, then uses an LLM to label and potentially merge the k-means clusters, thus discovering *an unknown number of* stable, human-interpretable clusters from unlabeled conversation data.

**Interaction Completeness**:
We fine-tune LLaMA-8B models on datasets of (mostly) successful conversations, tagged with a special "end" token, to model the distribution of each dataset. This allows us to predict whether a given interaction is complete, treating incomplete or failed interactions as outliers.

**Response Uncertainty**:
We propose the "response tree": a novel object constructed by branching LLM response generation to understand the distribution of possible responses. The response tree can quantify the LLM's uncertainty and provide interpretable insights into potential failure modes.

We empirically validate our metrics on LMSYS chat data and multiple agentic interaction datasets.
* Our LLM-guided clustering algorithm discovers interpretable clusters from data and is significantly more stable than a baseline LLM judge.
* Our approach to completion labeling, using LLaMA-8B models, matches or exceeds the performance of a 70B LLM judge.
* We provide a probabilistic argument relating the size of the response tree to the LLM’s response uncertainty.

## Summary of Reviews and Revisions
All four reviewers emphasized the novelty and applicability of our work to modern LLM systems. Reviewers’ concerns were focused on increasing the clarity of our paper and preparing additional results to validate our claims. In response, we made several important changes, which we summarize here.

**Unsupervised vs weakly supervised**:
Reviewers raised concerns about the “unsupervised” phrasing in the title due to our use of assumptions on the dataset quality. We **resolved this** by changing the title and wording throughout the paper to use the phrase “weakly supervised” instead.

**Verifying completion results**:
We computed p-values for the null hypothesis “our completion labeling does not have higher accuracy than an LLM judge”. We found **a statistically significant difference** on all datasets (except one where both methods are highly accurate).

**Verifying cluster stability**:
Reviewers suggested we use the Adjusted Rand Index, a measure of the agreement between two cluster assignments on the same dataset. We computed the ARI and found that **our algorithm is significantly more stable** than GPT-4.1 as an LLM judge. We updated Section 5.2 with these results.

**Justification for the response tree**:
We **added a probabilistic argument** in Section 4.3 relating the number of branches in the response tree to the likelihood of the LLM generating a particular (correct) response.

We also made various minor changes to improve the clarity of our paper. Overall, we believe these revisions have thoughtfully addressed reviewers' concerns and strengthened our presented results.

---

### Meta-Review · Area_Chair_oee4 · 2026-01-11

**Summary:**

The paper proposes a suite of weakly supervised metrics for evaluating multi-turn, objective-driven LLM interactions—common in enterprise agent scenarios—without human annotations or LLM-as-a-judge. It introduces three components: (1) LLM-guided clustering to discover user goals from unlabeled logs; (2) interaction completeness labeling via fine-tuned small LLMs trained on mostly-complete conversations; and (3) response trees to quantify model uncertainty by branching generation paths. Experiments across diverse datasets (e.g., LMSYS, WebShop, SQL+OS+KB) show that an 8B fine-tuned model matches or exceeds a 70B LLM judge in completion detection, and the clustering method is significantly more stable than GPT-4.1.

All reviewers recognize the strong practical motivation and novel evaluation paradigm, but raise concerns about statistical rigor, clarity of claims (e.g., “unsupervised” vs. weak supervision), and computational cost of response trees. The rebuttal thoroughly addresses these by adding p-values, Adjusted Rand Index for cluster stability, efficiency analyses, and clarifying assumptions. However, one reviewer (Xmjj) remains unconvinced, maintaining their reject score.

**Reviewer Concerns:**

Addressed by rebuttal:

“Unsupervised” misnomer: Authors renamed the approach “weakly supervised” throughout, acknowledging reliance on curated complete conversations.
Statistical significance: Added p-values showing completion labeling significantly outperforms LLM judges (except on one high-accuracy dataset).
Clustering stability: Reported Adjusted Rand Index, confirming LLM-guided clustering is significantly more stable than GPT-4.1 as judge.
Response tree justification: Added probabilistic argument linking tree size to response uncertainty and capped branching depth to control cost.
Efficiency: Clarified that fine-tuning + inference with 8B models is far cheaper than repeated 70B LLM judge calls, especially at scale.
Still outstanding:

Assumption fragility: Method assumes rare failures and single-goal interactions; performance degrades on Code-Feedback (multi-turn Q&A without clear end), highlighting limited robustness to task structure.
Reviewer Xmjj’s skepticism: Despite added evidence, this reviewer finds the rationale for metrics insufficient and maintains rejection—though others find the responses compelling.
Deployment feasibility: While response trees are theoretically sound, real-time use in long conversations remains costly despite pruning.

**Reviewer Scores:**

UJoU (initial: 6 – marginally above): Raised concerns on stats and clarity; all addressed with new analyses. Likely maintains 6 or moves to 7.

wVNW (initial: 6 – marginally above): Concerned about circularity and cost; rebuttal clarified assumptions and added stability metrics. Likely maintains 6.

hA7u (initial: 2 – reject): Criticized lack of empirical validation for response trees and batch-only clustering. Rebuttal added theoretical justification and ARI results, but core concerns partially remain. May upgrade to 4 or stay at 2.

Xmjj (initial: 2 – reject): Found rationale unconvincing; authors provided additional metrics, but reviewer explicitly stated they “did not find additional evidence” and maintains score at 2.

---

### Decision · Program_Chairs · 2026-01-26

Reject